# Exploring the Role of 6G Technology in Enhancing Quality of Experience for m-Health Multimedia Applications: A Comprehensive Survey [note 1]

**DOI:** 10.3390/s23135882

**Published:** 2023-06-25

**Authors:** Moustafa M. Nasralla, Sohaib Bin Altaf Khattak, Ikram Ur Rehman, Muddesar Iqbal

**Affiliations:** 1Smart Systems Engineering Laboratory, Department of Communications and Networks, College of Engineering, Prince Sultan University, Riyadh 11586, Saudi Arabia; skhattak@psu.edu.sa (S.B.A.K.); miqbal@psu.edu.sa (M.I.); 2School of Computing and Engineering, University of West London, London W5 5RF, UK; ikram.rehman@uwl.ac.uk

**Keywords:** e-health, healthcare, internet of things, m-health, quality of experience, 6G, multimedia

## Abstract

Mobile-health (m-health) is described as the application of medical sensors and mobile computing to the healthcare provision. While 5G networks can support a variety of m-health services, applications such as telesurgery, holographic communications, and augmented/virtual reality are already emphasizing their limitations. These limitations apply to both the Quality of Service (QoS) and the Quality of Experience (QoE). However, 6G mobile networks are predicted to proliferate over the next decade in order to solve these limitations, enabling high QoS and QoE. Currently, academia and industry are concentrating their efforts on the 6G network, which is expected to be the next major game-changer in the telecom industry and will significantly impact all other related verticals. The exponential growth of m-health multimedia traffic (e.g., audio, video, and images) creates additional challenges for service providers in delivering a suitable QoE to their customers. As QoS is insufficient to represent the expectations of m-health end-users, the QoE of the services is critical. In recent years, QoE has attracted considerable attention and has established itself as a critical component of network service and operation evaluation. This article aims to provide the first thorough survey on a promising research subject that exists at the intersection of two well-established domains, i.e., QoE and m-health, and is driven by the continuing efforts to define 6G. This survey, in particular, creates a link between these two seemingly distinct domains by identifying and discussing the role of 6G in m-health applications from a QoE viewpoint. We start by exploring the vital role of QoE in m-health multimedia transmission. Moreover, we examine how m-health and QoE have evolved over the cellular network’s generations and then shed light on several critical 6G technologies that are projected to enable future m-health services and improve QoE, including reconfigurable intelligent surfaces, extended radio communications, terahertz communications, enormous ultra-reliable and low-latency communications, and blockchain. In contrast to earlier survey papers on the subject, we present an in-depth assessment of the functions of 6G in a variety of anticipated m-health applications via QoE. Multiple 6G-enabled m-health multimedia applications are reviewed, and various use cases are illustrated to demonstrate how 6G-enabled m-health applications are transforming human life. Finally, we discuss some of the intriguing research challenges associated with burgeoning multimedia m-health applications.

## 1. Introduction

The healthcare industry has constantly adopted the latest technologies with the goal of improving the overall healthcare of the global population. The use of technology has significantly changed the way in which healthcare is provided [1], and it has had far-reaching impacts on delivering healthcare facilities to the general population, including those living in remote locations and in underdeveloped countries [2]. The advancements in technology have enabled the diagnosis of complicated diseases, and currently, sophisticated medical treatments are carried out easily. For instance, advancements in medical imaging technology have led to an increased understanding of the internals of human physiology, and this technology has become an important tool for diagnosing patient diseases, and conditions [3]. One of the most important applied technologies in healthcare is communications technology. The use of network and communication technologies has, over the decades, providing numerous benefits to healthcare. The earlier communication technologies, such as wired telephone lines, fax technology and networking technologies, supported quick and easy communication lines for the exchange of simple diagnostic information such as patient reports, treatment options, prescriptions, etc. The advent of wireless communication and the advancements in internet and satellite communication technologies have provided a significant boost to the delivery of healthcare [4]. They are necessary to revitalize primary healthcare since they contribute to the personalized monitoring of chronic diseases, increase access to healthcare for rural populations, and optimize data measurement and management. Patients must have to visit hospitals; it is generally a time-consuming and labor-intensive process that involves the participation of both the patient and healthcare workers. It often requires the management of several different healthcare aspects, making this process even more confusing and overwhelming. Hence, researchers turned towards smart healthcare using technologies such as wearable sensors enabled technologies, artificial intelligence (AI), and the Internet of Things (IoT) [1,5,6,7]. Moreover, the authors in [8] conducted a comprehensive review of E-Health Practices and Technologies from 2014 to 2019. E-Health refers to a range of technologies that utilize the internet to enhance the provision of healthcare services and improve the overall quality of life. Given the scarcity of research on this subject, their study employs a systematic literature review of articles published between 2014 and 2019 to identify prevalent e-health practices across the globe, along with the primary services offered, diseases targeted, and supporting technologies employed in this field.

### 1.1. Objectives

The objective of this survey paper is to provide a comprehensive analysis and understanding of the impact of 6G networks on user quality of experience management, particularly in the context of healthcare applications. The paper aims to bridge the gap in existing research by examining the relationship between 6G networks, Quality of Experience (QoE), and healthcare. The specific objectives of the paper are as follows:To extend the previous study by exploring the broader aspects of 6G networks in the field of smart health services and applications.To investigate how 6G networks can transform healthcare lifestyles and enable intelligent healthcare communication.To examine the advancements in cellular networks and technologies that contribute to the improvement of QoE in mobile health (m-health) multimedia applications.To demonstrate multiple 6G-enabled m-health multimedia applications and their impact on human life.To identify and discuss the research challenges associated with the development and deployment of multimedia m-health applications in the context of 6G networks.

### 1.2. Contributions

The literature on 6G networks presents numerous features, but there is a lack of research on managing user experience. It is believed that 6G will provide better and more personalized mobile experiences [9]. Advanced generations of networks demonstrated an evolution in terms of incorporating the user’s impression of the service. The consideration of network management was based on a QoS-based approach, where performance indicators were produced in terms of latency, jitter, and call drop percentage, among other metrics. However, combining voice with multimedia led to an expansion of evaluation to include a QoE perspective. This transition was prompted by a twofold developing reality at the time: network enhancements cannot be carried out without first analyzing the impact on user perception and without first acknowledging that elements other than network performance can influence user experience. QoS is concerned with network performance, but QoE is concerned with the performance as perceived by the users. Nowadays, network operators are placing greater emphasis on End-to-End (E2E) service quality in order to provide the best possible user experience. As a result, while migrating to 6G networks, it is critical that the developments in architecture maintain these quality criteria.

This research is an extension to our previous study in [10], which only looked at a few aspects related to 6G in smart health services and applications. In our previous study, we briefly identified and discussed the role of 6G in smart health services and applications, the critical 6G technologies that are projected to enable future smart health services applications, and the multiple 6G-enabled smart health applications. Moreover, there was not much elaboration in terms of taxonomy classification, diagrams, tabulations, discussion, recommendations and challenges, which could easily help researchers to build their future research in the field. On the other hand, to further improve the impact of this study, we have expanded the work in [10] to an extent where we look at how 6G could impact the future of healthcare lifestyles and demonstrate how 6G can be used for intelligent healthcare communications and specifically the QoE for m-health multimedia communications. We examine the advancements made in cellular networks to meet the demands of new m-health applications, such as high data rates, scalability, latency, and reliability. We shed light on several 6G technologies that are projected to enable future m-health services and improve QoE, including reconfigurable intelligent surfaces, extended radio communications, terahertz communications, enormous ultra-reliable and low-latency communications, and blockchain. Moreover, we demonstrate multiple 6G-enabled m-health multimedia applications along with various use-cases to show how 6G-enabled m-health applications are transforming human life. Finally, we discuss some of the intriguing research challenges associated with burgeoning multimedia m-health applications. To our knowledge, there is no publication in the literature where three major players, 6G, QoE, and healthcare, are discussed together. Table A1 provides a list of survey research papers and their primary focus; there are many surveys and tutorials that cover either one or two topics among 6G, QoE, and healthcare. Our survey is far broader, covering all three technologies.

Figure 1 provides the organization and systematic framework of the survey and is explained as follows. Section 2 discusses the theoretical background of m-health and QoE. Section 3 provides the methodology adopted in conducting this research and also discusses the findings and limitations. Section 4 provides an overview of QoE and the evolution of cellular generations. Section 5 covers the 6G features and advanced technologies that will transform the healthcare industry and significantly contribute to QoE enhancement of m-health multimedia applications. In Section 6, some 6G-enabled applications, such as holographic communication, haptic technology, AI, AR, and VR, have been investigated. Section 7 identifies various roadblocks to deploying 6G networks for healthcare, and finally, Section 8 concludes the paper. A list of key abbreviations used throughout the article can be seen in the end of the manuscript.

## 2. Theoretical Background

In this section, an overview of the different aspects used to facilitate the investigation of our research are provided. These aspects are Mobile-health (m-health), Multimedia Communications in m-health, Quality of Experience (QoE) of Multimedia Communications in m-health, and the Role of 6G towards Multimedia Communications.

### 2.1. m-Health

The rapid increase in mobile phone penetration across the globe has led to the advent of applications that deliver services in different areas, including healthcare. Several nations across the globe are adopting mobile-based technologies for healthcare. The World Health Organization (WHO), in their report on mobile-based technologies for healthcare, declared that over 80 percent of their member nations had made efforts towards the adoption and implementation of mobile technologies for health in their countries [11]. These technologies are popularly referred to as m-health technologies, where medical sensors and mobile computing for healthcare were the original definitions of mobile health (m-health) [12]. m-health works on the concept of any-time, anywhere access, exploiting the best connection features of emerging wireless communications and network technologies in a heterogeneous environment. Hence, it provides an alternative to the traditional healthcare delivery system and assists this system by bridging the gap between terrestrial boundaries and rural and urban areas. However, it is largely driven by successful business and market sectors. It still aims to reach the tipping point for clinical adoption and healthcare delivery efficiency. The three technology pillars of m-health, including telecommunications, computing, and medical sensing, drive this growing popularity. A massive number of smartphones and devices connected to a large number of mobile health applications are used worldwide by both patients and healthcare professionals. These services continuously generate structured and unstructured datasets, revealing new healthcare insights, but also creating some challenges. Today’s m-health apps are increasingly coupled to wearable sensors and Internet of Things (IoT) devices [6]. These applications are becoming common in everyday health, wellness, and clinical applications, including chronic disease tracking and management, remote patient data collection and monitoring, diagnostics and treatment, healthcare resource management, patients’ education and awareness as well as healthcare professionals’ training [13].

### 2.2. Multimedia Communications in m-Health

The different types of data transmitted via m-health applications could include text, image, and video data. In recent years, there has been a significant increase in medical imaging/videos for diagnostic purposes. Medical imaging technology advancements have resulted in portable scanning machines that have made producing medical images and videos easier. Subsequently, medical images and videos are also becoming an important part of the data transmitted via m-health applications [14]. Many use cases directly touch our lives and are now more complete and intuitive using multimedia technologies. Systems and services have been developed to use the benefits of multimedia technology, e.g., medical images and videos can be supplied to remote clinicians for consultation, and they can also be disseminated for educational and awareness purposes. As an added convenience in emergency situations, ambulances may be outfitted with portable scanners that capture and transmit diagnostic information to hospitals. Tele-consultation via video conferencing and remote patient monitoring sharing medical data can make individuals with chronic diseases see their doctors regularly and lessen the requirement for patient–physician visits. Gesture recognition and speech recognition are two technologies that are frequently used in conjunction with telemedicine in the smart medical home. Informing patients about diagnosis, surgery, and treatment is critical to their understanding of medical processes. Many patients are being educated using visual, audio, interactive, and internet content to supplement traditional paper-based information. As a result, medical video streaming is becoming an increasingly important component of m-health applications.

Multimedia services are time-sensitive, where efficient network systems are required to ensure service quality [15]. Due to the potential loss in quality in the video between the source and the destination, the quality of video transmission in m-health applications must be maintained at high levels so that the change in quality remains imperceptible. Further, medical videos carry sensitive data that determine patient diagnosis, and at times a loss in quality may impact the diagnostic understanding that can be gleaned from the medical videos. Thus, it is crucial that the quality of video transmission from source to destination is maintained at high levels in m-health applications.

### 2.3. QoE of Multimedia Communications in m-Health

To ensure that the quality of video transmission is maintained, a quality assessment needs to be carried out at the destination. Quality assessment is a process that employs methods to measure the quality of the video received at the destination compared to that of the video at the original source. There are two approaches to quality assessment; one is measuring the Quality of Service (QoS), and the other is Quality of Experience (QoE) [16]. In networking and communication applications, the QoS is a method of assessing the service quality that the network provides [17]. It is a network-centric measure that measures the quality of the data transmission provided by the communication network. The QoS measurement includes metrics that measure the key performance indicators (KPIs) of a communication system. These KPIs include metrics such as the percentage of data transmitted, the amount of data lost during transmission, channel throughput, speed of data transfer, system fairness, etc. On the other hand, QoE is a user-centric evaluation approach that reflects the degree of delight or annoyance of the user with the service used [18]. In the case of video transmission, the QoE of a user is the perception of the video quality by the end-user [19,20]. The QoE puts the focus of quality assessment on the user’s perspective and typically considers many factors surrounding the context of the application [21]. There are two broad categories of QoE evaluation: subjective and objective evaluations. Subjective evaluation methods are based on the Human Visual System (HVS), which largely depends on observation by human subjects. The subjective evaluation method uses an evaluation score known as the Mean Opinion Score (MOS), a mean score of the users’ opinions about their perception of the video’s quality. On the other hand, objective evaluation methods use mathematical models that act as quality metrics to measure the audio or video received quality, such as Peak Signal to Noise Ratio (PSNR), Mean Square Error (MSE), etc. A list of subjective and objective QoE assessment metrics are listed in Table 1 and Table 2, where more details and explanations about all these metrics can be found in [22,23]. Given that QoS primarily focuses on the technical elements, whereas QoE is centered around customer service and satisfaction, it is plausible to include QoS as an inherent component within the definition of QoE.

The QoS achieved in networks is now being increasingly associated with the QoE of the application. Recently, achieving high QoE has become an important part of multimedia communications [24]. The important features that affect the QoS and QoE in medical video streaming applications from the perspective of healthcare professionals include bandwidth, packet delay, jitter and packet loss tolerance, quality of the image, frame rate of the video received, real-time streaming when required, the direction of flow (uplink/downlink), mobility, etc. [25]. Measuring QoE can be complex, as several factors need to be considered that influence the end user’s QoE. The authors in [26] term these factors as “Influence Factors” (IF) and define them as: “Any characteristic of a user, system, service, application, or context whose actual state or setting may have an influence on the Quality of Experience for the user”. The m-QoE is the concept of traditional QoE seen from the perspective of healthcare applications. It is defined as the overall acceptability of m-health applications as perceived by end-users, where the end-users are the patients and healthcare professionals [27]. The m-QoE perceived in m-health systems is generally influenced by the system, context, and human influence factors (IFs).

### 2.4. The Role of 6G towards Multimedia Communications

Smart healthcare must be able to monitor health, diagnose diseases, and treat patients remotely with services such as Hospital-to-Home (H2H) services, emergency services, ambulance services, Intelligent Wearable Devices (IWD), etc. These services send the patient data remotely over the internet and then analyze them at data centers. Healthcare networks must provide mobility, high capacity, low latency (1 ms), green communication for patient safety, and continuous connectivity availability with high reliability. For these requirements, the sixth generation of wireless communication (6G) is regarded as the ideal solution for hosting healthcare networks [28,29]. Figure 2 depicts the healthcare network architecture in 6G; this diagram depicts the network’s key deployment tiers, including intelligent sensing, intelligent access, and intelligent cloud. This 6G architecture will enable many processes, such as security, flexibility, and intelligent resource management. This figure displays the slicing architecture of 6G networks, where different network slices enable different applications. The devices in the sensing layer gather data and deliver it to the next level. At this level, the nodes, users and devices connect to the radio access networks technologies such as massive MIMO, small cells, and advanced modulation techniques are implemented. Lastly, in the cloud layer, intelligent resource allocation, network management and optimization are performed. These different layers provide flexibility to the system on different levels, and a wide range of applications benefit from the network. This figure envisions such a scenario, demonstrating all layers and their key technologies.

The development of 6G communication technologies addresses some of the issues discussed in previous subsections, particularly in connectivity, data rates, network compatibility, and guaranteeing high levels of QoS and QoE for m-health applications [30,31,32]. A high data rate of at least 1 Tbps, extremely reliable, further-enhanced mobile broadband (FeMBB), ultra-reliable low-latency communication (URLLC), long-distance and high mobility communications (LDHMC), ultra-massive machine-type communications (umMTC), and extremely low-power communications (ELPC) are all features of 6G technology. QoS also incorporates parameters such as mobile broad bandwidth and low latency (mBBLL), and massive low latency machine type (mLLMT), revolutionizing many applications. The 6G technology must execute all desired characteristics to achieve good QoE. Thus, 6G technology will be groundbreaking in many fields as providing high QoS and QoE will enable new applications such as telesurgery, holographic communications, augmented and virtual reality, five-sense communications, and tactile internet. QoE will also play a vital role in intelligent transportation services and autonomous vehicles such as intelligent cars, drones, and ambulances [33]. One of the key goals of QoE in 6G is to allow service providers to assess the performance of their services and how specific service components contribute to that performance. From this viewpoint, we envision the research on the QoE of m-health multimedia to determine how various factors contribute to the QoE of the m-health application. Because of their nature, m-health applications differ from other applications. It is vital to determine how quality elements such as data, network, analytics, and so on affect the various stages of an m-health service and hence the service quality.

## 3. Methodology and Findings

This section of the paper aims to present the methodology employed for this comprehensive survey, and share the key findings obtained from the literature review. This section also discusses the limitations of the conducted literature review.

### 3.1. Methodology

The methodology of conducting this literature review can be divided into four steps, as shown in Figure 3, and described as follows.

Step 1 Research Paper Extraction: A comprehensive search was conducted using academic databases, including IEEE Xplore, Elsevier, MDPI, Springer, and other relevant sources. The search aimed to identify publications related to 6G networks, QoE, network management, m-health, healthcare, and user experience in mobile communications.Step 2 Research Paper Selection: The inclusion criteria for selecting publications included relevance to the research topic and publication date. Publications that focused on 6G networks, QoE evaluation, user perception, and network performance were considered. Publications that did not provide substantial insights into QOE, healthcare and 6G networks were excluded.Step 3 Paper Categorization: The papers were initially categorized as whether it is a survey/review or an experimental research article. The selected publications were thoroughly read and analyzed to extract key findings, theories, methodologies, and arguments related to QoE in 6G networks and m-health/healthcare.Step 4 Complete Paper Analysis: The extracted data were synthesized and analyzed to identify common themes, trends, and developments in the literature. Key concepts such as QoS, QoE, network management, and user experience were explored to understand their interrelationships and the evolution in the field. The analysis focused on the correlation of QoS and QoE, cellular network performance metrics, user perception factors, and advancements in cellular architecture, features of 6G, limitations of previous cellular generations in the context of emerging healthcare applications and requirements of next generation healthcare applications.

### 3.2. Findings

In this section, we present the key findings extracted from the literature review. These findings provide insights into the categorization of data based on major keywords, the distribution of cited papers by publishers, and the time series of cited publications.

#### 3.2.1. Categorization of the Cited Survey Papers Based on Major Keywords

As mentioned before, Table A1 in Appendix A lists the survey research papers with their primary focus. A summarized form of this data can be found in Table 3. We categorized these survey papers based on major keywords related to the research topic and the primary topic of discussion. The following categories emerged from the review:QoE Assessment: This category encompasses studies focusing on the assessment and evaluation of Quality of Experience (QoE) in 6G networks. These studies investigate metrics, methodologies, and frameworks for measuring and improving the user’s perception of network performance.6G Technologies: This category includes research papers that explore the advanced technologies and features of 6G networks. These technologies, such as reconfigurable intelligent surfaces, extended radio communications, terahertz communications, and blockchain, are projected to enable future m-health services and enhance QoE.Healthcare Applications: This category covers studies that investigate the application of 6G networks in the healthcare domain. It includes research on the use of 6G-enabled technologies for intelligent healthcare communications, m-health multimedia applications, and the transformation of healthcare lifestyles.QoE in the context of Healthcare: This category focuses specifically on studies that examine the Quality of Experience in the context of healthcare applications. These papers explore how QoE can be measured and improved in healthcare settings, considering the unique requirements and challenges of healthcare services.6G for Healthcare Applications: This category includes research papers that highlight the potential of 6G networks in enabling innovative healthcare applications. These studies explore how 6G technologies can be utilized to enhance healthcare services, improve patient outcomes, and enable new healthcare paradigms.QoE in the context of 6G: This category focuses on studies that investigate QoE in the context of 6G networks specifically. These papers explore the factors influencing user experience in 6G networks, the challenges in ensuring high QoE, and the strategies for optimizing QoE in 6G environments.

By categorizing the data based on these major keywords, we were able to identify the main themes and areas of research within the literature on managing user experience in 6G networks. This categorization provides a comprehensive overview of the different aspects and perspectives explored in the reviewed papers.

#### 3.2.2. Distribution of Cited Papers by Publishers

We analyzed the sources and publishers of the selected research papers. This distribution can be seen in Figure 4, where the search focused on existing literature on QoE, healthcare, 6G and related technologies. A significant proportion of the cited papers were published by the IEEE Xplore digital library and emerged as a prominent source of research papers on the topic. Several research papers included in the review were published by Elsevier, Springer and MDPI. The distribution of cited papers by publishers highlights the prominent sources of literature in this research area and the diverse range of publishers contributing to the knowledge base.

### 3.3. Limitations

In conducting this review, it is important to acknowledge the limitations of the research methodology employed. By understanding these limitations, readers can gain a better understanding of the scope and external validity of the conclusions drawn. The aim of this paper was to provide a comprehensive overview of the literature on QoE in 6G networks, particularly in the context of m-health applications.

Non-systematic Approach: Although efforts were made to collect papers from various sources, including IEEE Xplore, Elsevier, MDPI, Springer, and others, the lack of a formal systematic methodology might have resulted in potential omissions of relevant studies. As a result, the comprehensiveness and representativeness of the reviewed literature may be influenced.Selection Biases: The selection of papers was based on the relevance of their content to the research topic. While efforts were made to include a diverse range of perspectives, it is possible that some papers were unintentionally excluded, introducing a potential selection bias. Moreover, the availability and accessibility of certain papers might have influenced their inclusion, leading to a bias in the representation of the literature.

By acknowledging these limitations, we aim to provide readers with a transparent understanding of the potential biases and constraints associated with this narrative review. It is essential to recognize that while the conclusions drawn from this review are supported by the available literature, further research using systematic approaches and objective evaluations is warranted to provide more conclusive and refutable findings.

## 4. Overview: Quality of Experience and Cellular Generations

This section provides an overview of the relationship between Quality of Experience (QoE) and cellular generations in m-health applications. It addresses the limitations of 5G, the evolutionary progress of m-health wireless systems and quality of experience, and the correlation between QoS and QoE.

### 4.1. 5G Limitation for m-Health

Mobile technologies ranging from the first generation (1G) to the fifth generation (5G) have already been developed and deployed as an enabler for supporting a diverse range of networks and applications, including IoT. Notably, due to inherent usage characteristics such as eMBB, mMTC, and URLLC services, the 5G technology has been demonstrated to provide different service opportunities to IoT ecosystems with high throughput, low latency, and energy-efficient service provision [34]. However, 5G will be unable to fully address escalating technical demands such as autonomous, ultra-large-scale, highly dynamic, and fully intelligent services due to the unprecedented spread of smart devices and IoT networks. More importantly, as new IoT services and applications arise, such as remote robotic surgery and flying vehicles, further advances in current 5G systems are required to improve the quality of IoT service delivery and business [35]. The 6G wireless networks and the technological innovations that have accompanied them have recently gotten much attention from academia and industry and are expected to pave the way for the growth of the IoT and beyond [36,37]. Due to its superior features over previous network generations, such as ultra-low-latency communications, extremely high throughput, satellite-based services, massive and autonomous networks [38,39,40], 6G is expected to provide an entirely new level of service quality and improve the user’s experience in current IoT systems. In addition to enabling unprecedented capacity levels, 6G-based IoT networks will speed the development and implementation of IoT applications throughout the spectrum of data sensing, device connectivity, and wireless communication, among other areas. Figure 5 shows the evolution of cellular generations from 1G to 6G with their distinguishing features. This figure provides a synopsis of the evolution. It can be seen here how the last decade has brought a revolution in wireless communication by bringing applications such as multimedia, real-time communications, and high-definition video streaming. In the figure, we can see this evolution has resulted in various advanced services.

Although the 5G network has significantly improved the QoS over previous generations, it would still be difficult to fully meet the requirements of future m-health services [41], which are still in their infancy. The key performance indicators include parameters such as data rates, latency, network efficiency, coverage extension, reliability, spectrum efficiency, mobility and many more. The capabilities of 6G communication network as compared to its predecessor are shown in Figure 6, where the red indicates 5G and the blue as 6G. The 5G networks will be unable to handle the massive amount of mobile traffic generated by multimedia m-health applications due to higher screen resolution, machine-to-machine (M2M) communications, and mobile edge services. Because of the expansion of smartphones and tablets and the proliferation of mobile data services, the demand for mobile broadband (MBB) is increasing at a rapid pace. The services provided by 5G are insufficient to meet the evolving needs of healthcare technology and emergency situations. In this regard, 6G is envisioned to bring new disruptive wireless technologies and sophisticated networking infrastructures that would enable the realization of many new m-health applications by holistically addressing such rigorous network demands compared to 5G [42]. With the emergence of advanced technologies (e.g., edge intelligence, terahertz (THz), and large-scale satellite constellations), 6G communication systems will help the healthcare sector grow toward a more efficient, fully connected and intelligent system.

Billions of people throughout the world now use mobile and wireless networks. There will be a bottleneck for m-health applications due to the influx of traffic generated by all of these people being online simultaneously, which will put them at a competitive disadvantage with other high bandwidth-demanding services. As a result, wireless technology innovation is critical to keep pace with the ever-increasing demands for spectrum efficiency, energy consumption, mobility, flawless coverage, and the wide range of QoS and QoE needs. The network resources must be coordinated end-to-end and cross-layer to meet the needs and expectations of individual users [43]. A 6G mobile network is anticipated to provide QoS, QoE, and quality of physical experience (QoPE), as well as reliability and security [35,44]. Due to rate-reliability-latency and sensitive m-health applications such as telepresence, telesurgery, etc., QoE is a critical performance metric for 6G. The 5G QoS measurements, including packet loss rate, end-to-end latency, and round-trip timings, are no longer effective for upcoming 6G wireless applications. QoE focuses on user pleasure, aesthetics, comprehension, and application manageability. As a result, in addition to QoS, the outcome of usage is a critical component in ensuring QoE. Network and service management needs to be transformed by researching automation, cognitive operation, AI, and ML to increase QoE further. Quality is determined not only by network measurements in m-health multimedia applications, but also by other factors. Authors have focused on QoE in multimedia IoT apps in surveys such as [45,46,47], which is defined by how users utilize such applications. However, because 6G is still in its early stages, measuring performance metrics is susceptible to change at this time.

### 4.2. Evolution of m-Health Wireless Systems: 2G to 5G

Over the years and with the evolution of wireless systems, numerous m-health services and applications have been successfully deployed worldwide, satisfying the vision of pervasive connected healthcare. It is well-known that the evolution of wireless communication technologies has contributed significantly to the progression of m-health systems, starting from 2G (GPRS) and 3G (UMTS, CDMA) through WiMAX to the existing 4G (LTE/LTE-A), current 5G and towards future 6G systems [27,48]. m-health services are beneficial in areas with a shortage of medical health facilities, where hospitals are located far from large populations, or where medical care is costly. Satellite communication, cloud-based systems, and 4G and 5G cellular networks are all examples of innovative and advanced wireless and wired technologies used in m-health systems. Technology is altering traditional healthcare services and saving lives by providing rapid diagnosis and alerting medical centers, such as advanced cellular technologies, smart small wearable gadgets that monitor and communicate data to patients’ and medical assistants’ smartphones [49,50].

While security, dependability, latency, and physical size are all critical considerations for m-health systems, one of the most significant barriers to the broad adoption of cellular networks in telemedical systems is a lack of bandwidth and performance [51]. The first generation (1G) of analogue mobile phones aimed solely at voice communication. The 2G Global System for Mobile Communications (GSMC) phones were designed primarily for digital voice transmission (GSM). General Packet Radio Service (GPRS) is offered as a network overlay for GSM, Code Division Multiple Access (CDMA), and Time Division Multiple Access (TDMA) networks formed 2.5G. The initial GSM service only allowed for data rates up to 9.6 kbps per channel, and GPRS allowed approximately 14 kbps per channel. In theory, GPRS can combine up to eight channels, bringing the total bandwidth to 100 kbps. For the first time, the primary objective of wireless phone service was data transmission rather than voice transmission. For end-users, the most crucial characteristic of GPRS was that it provided constant connectivity. Various services were made available over GPRS, including web browsing, the transfer of still photographs and video clips, document sharing, and remote collaborative working. These services contributed to the development of the world’s first usable medical systems, which were built on global accessibility. Enhanced Data for GSM Evolution (EDGE) was denoted by 2.75G. If the GPRS protocol was the first step on the road to 3G, the EDGE protocol was the next step on the road. In theory, EDGE provided data rates of up to 384 kbps, while actual data speeds were lower. The ability of EDGE to operate on the current GSM spectrum was the main attraction of the technology.

Increased data rate requirements and technological advancements necessitated the advent of 3G technologies, with the primary air interface being wideband CDMA (WCDMA). This generation was named the Universal Mobile Telecommunications System (UMTS). UMTS supported data transfer rates of up to 2 Mbps indoors (a low-mobility scenario), 384 kbps outdoors (at the speed of slow-moving pedestrians), and 144 kbps for fast-moving mobile phones. The overall goal of UMTS was to provide seamless functioning through the employment of pico- and microcells indoors and in urban areas, macrocells outside and in rural regions, and, when necessary, satellite networks. When the 3G systems were being defined in their infancy, video telephony was envisioned as a breakthrough application. The 3G systems have been demonstrated to efficiently support the amount of data necessary for medical applications [52]. The emergence of 3G wireless systems provided an initial platform for transmitting medical images and videos. The requirements after 3G started increasing rapidly as new systems required more bandwidth, steady performance, and a high QoS.

The introduction of 4G technology was a turning point for m-health applications as it overcame the limitations of previous wireless communications generations. The amalgamation of 4G technology with healthcare brought in the concept of 4G health, which can be defined as m-health using personalized medical systems with adaptable functionalities and compatibility with future 4G network technologies. In the 4G systems, the High-Speed Downlink Packet Access (HSDPA) protocol supported data rates of up to 14.4 Mbps with a latency of around 100 ms. The 5G mobile communication technologies and their related IoT-connected ecosystem served as a catalyst for the development of m-health applications. Mobile health apps benefited from the unparalleled speed, capacity, and low latency levels provided by these technologies. Data rates of up to 10 Gbps are possible, representing a 10- to 100-fold boost over the 4G networks with one millisecond or less latency, more than 100 times the number of connected devices per square meter compared to its predecessor. Network coverage levels exceed 100 percent while maintaining a fairly constant network availability [12].

### 4.3. Evolution of Quality of Experience: 2G to 5G

QoS has traditionally been used to evaluate a system’s overall performance in terms of its service to its users. The intent of the QoS was not to capture the subjective perception of quality or level of pleasure held by the user. The International Telecommunication Union (ITU) first proposed the concept of QoS as experienced by the user in 2008 [53], stating that it is “influenced by the delivered QoS and psychological factors influencing the perception of the user”. As a result of the significant advances in information and communications technology, the term Quality of Experience (QoE) was coined at a later stage. According to [53], QoE is the total acceptability of an application or service as assessed subjectively by the end-user. As a result, the user-centric idea of QoE complements the notion of network-centric QoS. The primary goal of the QoE program is to assess customer perceptions of quality and collect feedback to support the optimization of various technological aspects [33]. Voice as the primary service in the first two wireless generations, measuring user impressions relied on mean opinion score (MOS)—a subjective measure widely employed in traditional phone networks. GSM enabled a significant increase in mobile data rates leading to the GPRS and the EDGE that enabled the first widespread adoption of Multimedia Messaging Services (MMS). It enabled multimedia service applications to prosper and provided increased speed, bandwidth, security, and mobility management while improving radio resource utilization [54]. UMTS defined the QoS requirements for multimedia wireless data servicesand necessitated the development of a new architecture capable of reducing the latency (100 ms) using end-to-end IP data communication. Additionally, 4G and LTE-A enabled improved mobile multimedia functions and increased user equipment flexibility. The LTE-A enabled the efficient sharing of multimedia resources across the eNodeB, the Evolved Packet Core (EPC), and the Content Provider via the Multimedia Broadcast Multicast Services (eMBMS) pioneered a new business model based on the LTE radio architecture. Additionally, by implementing the Mobility Management Entity (MME) on activated eMBMS on 4G networks, LTE-Broadcasting (LTE-B) was enabled, conclusively enabling the simultaneous distribution of multimedia resources to several users at the same time, thereby enhancing QoS. The MBMS Signal Frequency Network (MBSFM) converts multimedia content sent from several eNodeBs to a multipath signal to increase the gain of the original content that is being multicast or broadcast. The eMBMS bearer contributes immensely to video compression methods and new multimedia technologies. eMBMS leverages MIMO technology to provide consumers with Multicast/Broadcast media content that supports a high level of QoE, such as High-Definition Video streaming.

The 5G NR introduced new functionalities to improve QoE by merging the concepts of Network Slicing, Edge Computing, and the 5G QoS Flow. With the introduction of new features such as improved Mobile Broadband (eMBB), Ultra-Reliable Low Latency Communications (URLLC), and massive Machine Type Communications (mMTC), new opportunities for service applications have opened up. Numerous m-health multimedia apps were made possible by these components. When considered in light of the ongoing discussions on 6G networks, QoE considerations take on new dimensions that have not been considered previously, particularly in light of the projected enhancement of the user context through augmented reality (AR) and virtual reality (VR), in addition to tactile experiences. It becomes vital, as a result, to streamline the QoE investigation toward relevant and usable procedures.

### 4.4. The Correlation between QoS and QoE in m-Health Applications

In medical video streaming, especially for real-time diagnosis, it is crucial to have a highly reliable network that is predictable and performs as required, even in varying conditions. Furthermore, the wireless network should be able to support real-time traffic management for the medical video streaming application. Traffic management is important because real-time traffic tends to be bursty in nature, and this may often lead to packet losses, which in turn implies the loss of information; hence, the video packet streams need to be well-managed. The loss of packets is not acceptable for medical applications because it may hinder the accurate reconstruction of the received videos, which may further affect the diagnosis process. Therefore, a medical video streaming platform must be able to support a maximum, average, and minimum set of traffic parameters.

The m-QoS aspects related to m-health applications have been investigated extensively in the literature. However, the m-QoE aspects are not yet well-explored. A challenge in m-QoE is that users generally have higher expectations in terms of quality received as the service is related to their health, and they are vulnerable to losing trust in the service provided by the m-health system. Therefore, in order to achieve high acceptability of m-health applications, the m-QoE aspects need to be determined and incorporated into the design. To determine the context of the m-health application, it is important to identify and analyze the clinical-related requirements and the influencing factors such as application area, application purpose, content type, and context of use.

6G Technology promises to deliver high QoS [36,38], enabling a plethora of new applications making 6G technology a game-changer in numerous fields. Additionally, 6G technology promises high QoE, which defines user-centric communications. m-health multimedia communications refer to clear and detailed signals, which can be audio, video, bio-signals, text, graphics, still images, interactive data, and so on. The optimization of the QoE during medical media transmission is one of the most significant concepts to consider from the perspective of the end-users perception [55].

Many studies have focused on determining the relationship between QoS parameters and QoE; few instances claim that QoS is the most critical factor in determining a user’s perceived quality [56]. Correlation models must be established to evaluate the relationship between QoS parameters and their individual and mutual effects on QoE [22,57]. The correlation models proposed in the literature can be divided into general and specific models [58]. The prior one is shown in Figure 7. This figure talks about the general model of correlation between QoE and QoS. It considers QoS as a component in a series of QoE-determining variables and shows that QoE cannot be separated from QoS metrics as the network’s state influences the user experience. Figure 7 shows where QoS is specific to the transmission, while QoE includes transmission as well as involves the end-user’s engagement. The figure illustrates prospective application scenarios of m-health, such as intelligent ambulance services, remote patient monitoring, telemedicine, etc.

Alternatively, specific models investigate the impact of QoS parameters on QoE. QoS/QoE mapping calculates QoE values based on a set of measurable network KPIs from the QoS objective criteria, referring to the degree of service adequacy. Using appropriate mathematical models, QoE can be obtained from these measurements. Subjective user-centric aspects, which cannot be measured directly from the network but may impact the end-entire user’s experience, are included in the QoE influencing factors [59]. Understanding the relationship between network-based QoS parameters and end-user-perceived QoE is crucial in the QoE management process [60].

QoE cannot be separated from QoS metrics as the network’s state influences the user experience [9]. However, the tendency to view the network as the primary element in determining QoE should be addressed. Research has been conducted on finding QoS parameters affecting QoE in telecommunication systems [61,62]; for example, the authors in [63] show how QoE has a logarithmic relationship with the throughput in transmission. Studies such as [64] improve the modeling of QoE–QoS interactions, but QoE consideration is still limited to what can be measured and managed through the network. The researchers in [65] classify multimedia as a key performance indicator for QoE and QoS analysis. A reliable and efficient technique for analyzing QoE is designed to determine the trade-off between user perception and QoS entities [66]. Additionally, they addressed the fact that standard network performance metrics such as delay, jitter, and throughput do not accurately reflect the complete user perception level. However, by combining QoE and QoS parameters, the desired result can be attained. Similarly, researchers in [67] provide a mechanism for energy conservation and QoS evaluation in healthcare and traditional wireless networks.

Overall, 6G is planned to enable a wide range of m-health applications, making it more challenging to handle the unique QoS requirements of different apps while also ensuring the QoE of different users. The ability to derive complicated quantitative correlations between QoS parameters and KPIs is crucial for controlling QoS. In order to do this, Ref. [68] suggested a delay estimation approach based on neural networks. The input to the supervised learning technique is the traffic load and the routing strategy, while the output of the supervised learning approach is the network delay. According to the findings of the experiments, the NN-based estimator outperforms the standard M/M/1 model in terms of estimation accuracy by a significant margin. Similar to this, Ref. [69] attempted to uncover the mapping relationships between QoS parameters and KPIs through a decision tree. The linear regression technique M5Rules is used to determine the quantitative influence of KPIs on QoS. Obtaining the QoE values takes a significant amount of time, and as a result, understanding how various QoS parameters affect the QoE values is particularly crucial when computing QoE values quickly. The goal of [70] is to measure the QoE of a video streaming service using a supervised learning approach. Network metrics such as RTT, jitter, bandwidth, and delay are utilized as input to train the NN model, while the QoE is employed as an output. The SDN controller can adjust network parameters to increase QoE; on the other hand, Abar et al. [71] attempted to estimate QoE values from video quality characteristics. Four machine learning algorithms are investigated: the decision tree, the neural network, the K-NN, and the random forest. By applying MLP, Pierucci et al. [72] studied the relationship between user throughput and QoS factors to forecast users’ QoE based on user throughput, the number of active users, and channel quality indicators, among other things. The simulation results reveal a high degree of accuracy in making predictions. As new applications, such as XR in 6G, have emerged, new QoE and QoS evaluation metrics are introduced. Existing research reveals that deep learning is promising in learning the complex relationships between them, allowing for more efficient QoE insurance and resource utilization.

According to the authors in [73], QoE is a broader concept that incorporates QoS metrics and user-related elements such as their behavioral, cognitive, and psychological states and the context in which these products and services are offered. These elements influence the overall QoE when using a particular application or service. Mobile computing environments are characterized by dynamic user behavior, as users interact with applications or services in various situations. Consequently, when measuring and predicting QoE, it is critical to take into account metrics relating to users, their device(s), and the surrounding environment, as well as network QoS. User moods, screen size, social context, activities, and packet loss, are just a few examples of the types of information that can be collected and stored.

## 5. 6G Features Significantly Enhancing the QoE of m-Health Multi-Media Applications

As a result of the COVID-19 pandemic, internet video streaming has already taken over a major portion of our daily lives through various applications such as webinars and video conferencing. However, issues such as video startup failures, pixilation, delays, or other sorts of image or audio quality degradation were commonly seen during the quarantine period in all parts of the world. All of these degradations are unacceptable in the context of m-health services. The need to deal with the explosion of multimedia services has been considered in the 6G network, which will provide greater QoS while also guaranteeing QoE.

In this section, we provide greater discussions about 6G features which specifically discusses the following: the provision of enhanced video streaming using small cells solutions, the flexibility of 6G high frequencies, the phenomena of the 3CLS services, the enhanced multimedia transmission services, the provision of network slicing and artificial intelligence, enhanced security and blockchain, reconfigurable intelligent surfaces, extended radio, and finally the massive ultra-reliable and low-latency communications.

### 5.1. How Small/Tiny Cells Can Enhance the Video Streaming

The deployment of small cells in an m-health wireless infrastructure would mean that patients and healthcare professionals on handheld devices would experience high QoS and QoE down at the street level, where it really matters. The expansion of video traffic has increased the strain on spectral efficiency, energy consumption, mobility, seamless coverage, and the various requirements for QoS and QoE. Medical multimedia applications will coexist and compete with conventional video traffic to deliver healthcare services across mobile networks in a shared environment. To address the aforementioned issues, the adoption of tiny cell (e.g., small cell) technology as a communication platform for strong m-health systems is recommended.

Small cells (e.g., femtocells) are low-power base stations (BSs) specifically designed for coverage enhancement in an indoor dense environment where there is limited macrocell network coverage. They are characterized by the low transmission power, high capacity, low cost and easily deployable features with a coverage range of some tens of meters. They can be readily installed by the users in a plug-and-play manner, with an option of restricting access to only a certain group of users present at home or in an office environment (i.e., closed access). They perform the entire operational task in a completely autonomic and self-organized manner, including device configuration, troubleshooting, and optimization [74]. Due to the limited number of users permitted access to the small cell BS, each user can achieve high data rates. The demand for improved data rates along with spectral and energy efficiency at any time in 6G motivates the exploration of frequencies above sub-6 GHz. As 6G evolves, it will become essential to use frequencies beyond mmWave, in the terahertz (THz) range (i.e., referring to electromagnetic waves with frequencies between 100 GHz and 10 THz) [75]. To take advantage of higher mmWave and THz frequencies, the 6G cell size must be reduced from small to “tiny cell” with a radius of only a few tens of meters. Small cells (e.g., femtocells and picocells) have a different radio range, and network characteristics, which enables them to transmit at a higher rate while reducing traffic on macrocells [76].

### 5.2. 6G Supports Higher Frequencies

As discussed previously, 6G enables higher frequencies, which is advantageous for multimedia applications requiring high data rates. It also provides spectral and energy efficiency, high mobility, and seamless coverage. Additionally, 6G will require the continuance of Massive-MIMO technologies to offer high data rates efficiently; frequencies above the radio spectrum (i.e., Terahertz) are strong candidates for use with Massive MIMO [35,54]. THz communications are envisioned as the driving technology for 6G-IoT, demanding data rates of 100+ Gbps and latency of less than one millisecond. Receiving high data rates to the order of Gbps would let healthcare professionals view high-resolution medical images and videos in Ultra-High-Definition format. The THz band (0.1–10 THz) appears to be a good candidate for future m-health needs, including picosecond-level symbol duration, integration of thousands of submillimeter-long antennas, and low interference without requiring complete legacy regulation [77]. Some benefits of the THz spectrum for 6G-enabled m-health systems include addressing spectrum scarcity issues in wireless communications, allowing ultra-high bandwidth and throughput, and enabling ultra-broadband applications such as virtual reality and Wireless Personal Area Networks (WPAN) to be implemented [78].

In m-health wireless systems, the scalability of multiple medical sensors/devices and seamless connectivity are requirements that future 6G technology is anticipated to support. Body wearables and implants involve multiple biomedical/nano sensors and actuators that take physiological readings from the human body for m-health remote monitoring applications. 6G THz frequency bands can accomplish Internet of Nano-Things (IoNT) communication [68], where densely deployed nano-things continuously generate valuable data and forward it to data centers over a high-speed 6G communication network [79]. Similarly, the Tbps data rate needs for AR/VR services can be supported via THz transmission and Tiny cells [80]. Given the significant free-space path loss associated with THz signals, including spreading and molecule absorption loss [81], the THz bands are better suited for short-range wireless communications, such as wireless backhaul for tiny cells. THz communication will evolve steadily over the next few years and will be critical in the 6G sector, with numerous promising technologies [82].

### 5.3. 3CLS Services

Overall, 6G will significantly expand the wireless network depth from single information transmission to information transmission, storage, and processing, which helps in maximizing network utility and providing exceptional QoE for various services and applications. As a result, sophisticated wireless sensing, communication, processing, caching, and control must all work together. With this convergence, the networks can make optimal decisions related to sensing and computing tasks [45]. In the previous five generations of cellular systems, only the feature of wireless communication was available, but 6G will provide a variety of services through the convergence of Communication, Computing, Control, Localization, and Sensing (3CLS) [35]. As a multi-purpose system, 6G has the potential to provide many 3CLS services, which are particularly interesting and even critical for applications such as extended reality (XR) services (encompassing augmented, mixed, and virtual reality (AR/MR/VR)), Connected Robotics and Autonomous Systems (CRAS), and Distributed Ledger Technologies (DLT) that require tracking and control, as well as localization, computation, and other features.

Using the same time-frequency-spatial resources, communication, location, and sensing services are all expected to coexist in 6G networks [83]. This coexistence will significantly improve spectrum efficiency while also avoiding present regulatory obstacles. The cooperation between these services has significant mutual benefits [84], e.g., location and context-aware communication. This context awareness will also assist m-health applications in 6G networks by allowing multi-modal communication and service quality based on the present location or context. For example, based on the application needs, one can select a subset of terrestrial, aerial, and space elements using accurate context awareness [85]. Applications such as Human-Centric Services (HCS) are predicted to rely on 3CLS services to provide effective communication and real-time processing of a huge number of data streams collected by sensors centered on humans [86]. Existing 5G technologies have not thoroughly investigated the interdependence between 3CLS in an end-to-end manner; thus, the development of 3CLS services is an essential driving trend toward the next generation of mobile communication networks [87]. Future mobile communication networks will need collective network intelligence at the network’s edge to run AI and ML algorithms in real-time to provide 3CLS services [35]. In order to support AI-managed end-to-end 3CLS design services, the network architecture should also be open, scalable, and elastic [35,88].

### 5.4. Evolved Multimedia Broadcast Multicast Services

Current cellular IoT systems monitor service announcements even when firmware and software changes are infrequent. The 6G IoT devices would no longer need to watch service announcements but instead be paged to obtain multicast data resulting in a reduction in energy consumption [89]. The Evolved Multimedia Broadcast Multicast Services (eMBMS) bearer is a valuable addition to video compression protocols and new technologies devoted to video applications. It uses MIMO technology to deliver Multicast/Broadcast media material to users while maintaining a high level of QoE, for example, high-definition video streaming [54].

Mobile cellular networks offer great potential to support vehicular services due to their high network capacity and widespread coverage, but still some challenges persist. Vehicular data should be routed through the base station (BS) first, as centralized cellular networks have limited applicability for vehicle-to-vehicle communications [90]. In unicast mode, a vehicle transmits emergency information to a cellular BS, then unicasts it to the appropriate vehicles; this limits the downlink channel. MBMS and eMBMS are viable methods for disseminating emergency information [11]; eMBMS is typically used in conjunction with a single-frequency network, where multiple BSs broadcast simultaneously over the same channel [12]. Multiple vehicles can acquire emergency information concurrently utilizing the same frequency band. When multimedia content is sent from separate eNodeBs, the MBMS Signal Frequency Network (MBSFM) converts it into a multipath signal, which can then be used to improve the gain of the original content, that is multicast or disseminated.

### 5.5. Network Slicing

User demands for high-quality services have prompted telecom operators to invest in new cutting-edge network softwarization paradigms such as Software Defined Networking (SDN), Network Function Virtualization (NFV), Multi-Access Edge Computing (MEC), and Cloud/Fog Computing (C/FoC). Pressure from new emergent use cases such as ultra-high quality video (4K/8K), network-controlled D2D communications, Machine Type Communication (MTC), and Massive Internet of Things (MIOT) is also driving this transformation and system upgrading. The future networks will be cloudified via SDN and NFV, enabling Internet Service Providers to customize services and provide QoE that matches consumer needs through sophisticated QoE control and management [45].

Virtualization is a technology that provides virtual representations of things such as virtual computers, servers, memory, network services, etc. The NFV allows network functions to be separated from proprietary hardware appliances and run in software on standardized hardware. SDN adds intelligence and flexibility to programmable networks, allowing finer-grained network-wide orchestration and management of operating applications and services [91]. SDNs can have their QoE control and management planes built as software that runs on commodity hardware devices [45]. SDN and NFV aim to establish a future software-based networking solution that enables end-users to have flexible and automated network connectivity and QoE provisioning. The use of AI in conjunction with SDN/NFV/NS allows for dynamic and zero-touch network orchestration, optimization, and management, accelerating the transition from 5G to autonomous 6G networks. AI-assisted network optimization can track real-time network KPIs and make rapid adjustments to network parameters to maintain high QoE.

Different strategies for QoE-centered management of multimedia streaming services are applied using SDN/NFV technologies. Maximum QoE for end-users is achieved through the best path calculations for each service flow. The video is broadcast through low-latency, high-throughput networks, which improves the attained video quality [92]. The computation of optimal bitrates for clients to achieve QoE-fairness in the delivered video is performed by assigning different priorities to a video stream depending on the requested quality [93,94]. Allocation of network resources cost-effectively ensures that the highest possible level of user QoE fairness is attained. The underlying concept is to prevent quality instability, unfair bandwidth sharing, and network resource underutilization among DASH competing clients who share the same bottleneck network link to maximize network usage [95,96]. Coordination of actions and cooperative cooperation between layers are required to optimize network resources and improve the end-user’s QoE. Using SDN controllers, the QoE requirements can be set at the application layer and regulated at the network layer in such a configuration [97]. In SDN-based networks, it is necessary to support effective multipath routing and accelerate the transmission of huge amounts of multimedia applications between end points while maintaining the QoE of end-users. The MultiPath TCP (MPTCP) enables load balancing, reliable communication, and improved network resource usage, which results in increased network throughput and improved QoE for end-users [98].

### 5.6. Artificial Intelligence

QoE models based on Machine Learning (ML) and Artificial Intelligence (AI) that consider video features can be extremely useful for m-health multimedia applications. AI-assisted network optimization can track real-time network KPIs and make rapid adjustments to network parameters to maintain high QoE [36]. “Pervasive intelligence” is proposed as one of the 6G network’s ultimate goals, which aids in the formation of an auto-generating and auto-configuration network paradigm [99]. The network is designed to configure processing algorithms, signaling autonomously, and protocol processes for future new use cases and more complex requirements, potentially reducing standardization efforts and costs in implementing new network infrastructures.

AI-enabled RAN design is proposed in [100] to minimize the cost of standardization while maximizing the potential gain of AI-enabled network optimization. One AI scheduler controls all tasks of the RAN, including RRC, RRM, protocol function elements (PFEs) setting, and MAC layer scheduling. Deep learning, reinforcement learning, Q-learning, and other AI approaches can be used to learn the characteristics of the services [101]. The scheduler might be more efficient and intelligent based on the prospective prediction of channels, data traffic, QoS/QoE indicators, and so on at either the AI controller or AI scheduler. The non-real-time AI processing is handled by the RAN AI controller, which has logical interfaces with the CN, network operations, administration, and maintenance (OAM) platform, and BS. The AI controller may forecast user mobility, traffic behavior, network loading fluctuation, and other factors using big data analytics [102], which is then supplied to the BS to assist the AI scheduler with RRC, RRM, PFE orchestration, and real-time MAC scheduling.

The AI controller and scheduler could use deep learning approaches based on historical and new data to figure out customized RRC and RRM functions set for a certain UE in a specific circumstance. Instead of regular dynamic scheduling, the network can allocate a reserved semi-static resource, such as semi-persistent scheduling (SPS) and configurable grant (CG) configuration(s), to a device based on its traffic pattern acquired from application-layer data. A service’s unique requirements for dependability and survival time might also be supplied, which would aid the scheduler in determining whether to use a duplication mechanism or repeat transmission for devices.

### 5.7. Enhanced Security and Blockchain

Achieving a high level of security and privacy in 6G-based IoT networks is a real challenge since 6G systems are distributed and, as a result, face higher risks of cyberattacks and threats. Additionally, maintaining data privacy in multilayer 6G open-sharing systems such as vehicular data sharing in automated vehicles is a crucial issue. In terms of security attacks, signals used for dedicated short-range communication in autonomous ambulances can be altered or controlled to jam the radar or cause collisions with other vehicles. Aside from that, smart wristbands are worn by restricted patients to monitor and track their movements around the hospital facility. However, the communication signals transmitted by wristbands are susceptible to jamming and spoofing attempts. Overall, 6G will bring a new solution to the security paradigm, redefining it entirely. Blockchain [103], as an emerging disruptive technology, can provide novel solutions to such privacy and security concerns in 6G-IoT networks. The blockchain, in theory, is a decentralized, immutable, and transparent database in which no authority is required to administer the data. It is made possible by a peer-to-peer network structure, which gives each entity, such as an IoT device, an equal right to control and authorize the data recorded in the blockchain.

Due to the sensitive nature of healthcare data, blockchain adoption is required to ensure utilitarianism and other normative issues of rights and duties [104]. There are significant ethical concerns as this sector deals with individually identifiable and confidential health-related data. The verification of health data in 6G-based healthcare systems can be accomplished using blockchain and its inherent smart contract technology [105], eliminating the need for a third party while maintaining a high confidence level. The operational reach of blockchain-enabled systems may span multiple jurisdictions with varying medical and cyber-security requirements. So, if any blockchain-enabled healthcare application is to be applied across national or regional borders and provided via a high-speed network, standardization, transparency, and accountability are required [104,106]. Apart from blockchain, federated AI, quantum machine learning, quantum computing, and THz communication will be used to fight against cyberattacks in 6G [107,108]. THz communication is impenetrable to eavesdropping and jamming.

Spectrum sharing is an expensive feature in 6G or lower cellular networks because large quantities of money are paid to spectrum regulators. The operator purchases bands and sub-bands and then uses or leases to other operators. Blockchain technology can be utilized for license shared access (LSA), and cognitive radio networks (CRN) can be used with blockchain to create a distributed MAC protocol for secondary network users [109]. Furthermore, spectrum sharing based on blockchain would enable network resource virtualization. The federal communications commission (FCC) has stated that because blockchain is an important component of 6G stacking applications, a blockchain-based spectrum-sharing service can be implemented in the 6G protocol stack [110].

### 5.8. Reconfigurable Intelligent Surfaces

Overall, 6G will enable a plethora of new possibilities and state-of-the-art applications requiring minimal latency and high-quality video and huge image data transmission. Reconfigurable Intelligent Surfaces (RIS) and orbital angular momentum communication could serve as core technologies for the 6G radio interface, providing a solution to many physical layer constraints [9]. This new paradigm of programmable and software-defined metasurfaces will offer previously unimaginable technical solutions for unobtrusive and noninvasive sensing, which has the potential to revolutionize the healthcare industry [111]. Recently, RISs have garnered significant research attention for applications, including 6G technology [112]. The RIS comprises arrays of passive scattering devices with artificial planar structures. Each element is enabled to reflect the impinging electromagnetic wave in a software-defined manner using electronic circuits [113]. Due to their reconfigurability, RISs are a possible solution for assisting in the design and optimization of wireless systems by facilitating signal propagation, channel modeling, and acquisition, hence enabling smart radio environments advantageous for 6G-based applications.

This integration of RIS enabled-Intelligent Walls (IW) will play a critical role in the development of the 6G communication ecosystem. Diverse factors necessitate seamless coverage and connectivity in healthcare settings. The hospital’s complicated physical structure and sensitive equipment restrict network coverage. Currently, the hospital uses Wi-Fi networks for communication, but electromagnetic interference limits coverage in sensitive places [114]. In order to improve coverage and safeguard RF-prohibited areas, RIS can be installed on walls, ceilings, etc., in hospitals [115].

Interconnected medical equipment and applications capture data and send it to healthcare IT systems. Wearable healthcare devices must be charged periodically to stay active and transmit patient data to monitoring systems. The IW will be an indoor primary surface used to charge wearable gadgets wirelessly [116]. The IW’s sensor circuit would listen for the wearable device’s beacon signal and direct the ambient RF energy toward it when a battery threshold was met. Nevertheless, most wearable gadgets use backscatter communication, which requires a separate transmitter nearby. Wearable devices cannot transfer data over extended distances due to power limits and limited data rate transmission. Thus, in IoMT applications, IW can be employed for both charging and rapid data transfer. Noninvasive patient sensing is another case in healthcare [116,117]. The RF signals could be utilized as an illuminator of opportunity, combining an ambient signal from a BS with echoes from test subjects to collect information on their movements [118]. Using ambient signals, it can obtain high-resolution full-scene images and accurately detect human body postures and vital signs [116,119]. The AI-enabled controller attached to the IWs can process the data. A patient with heart attack symptoms or an elderly patient who falls over would order a secondary IW to redirect the ambient mm-wave signal to the specialized emergency response radio link. RIS can also provide great benefits for services such as localization and mapping [120].

### 5.9. Extended Radio (XR)

Overall, 6G will be supported by Extended Radio (XR) by having satellite communication in conjunction with the terrestrial connection. The 6G technology’s primary needs are a 1 THz operating frequency, and a 1 Tbps data throughput [108]. Existing 5G technology is primarily focused on metropolitan areas, with less emphasis on developing and rural areas. Thus, a broad swath of the population area is covered with deep indoor penetration, ensuring adequate equity for rural areas. Overall, 6G will provide an interconnected communication network ranging from the ground and sky to the underwater environment and satellite communication to achieve global connectivity. Unlike today’s IoT deployment, primarily based on terrestrial infrastructure, 6G communication utilizes diverse low and high-altitude platforms, including Low Earth Orbit (LEO) satellites, to form an aerial network framework that supports ultra-massive IoT connections via reliable wireless links. The aerial network is also supported by UAVs and balloons that act as flying BSs, providing coverage and connectivity for disaster-affected areas and supporting public safety networks and emergency scenarios requiring URLLC [121]. UAVs integrate with terrestrial BSs to provide direct air-ground links for collaborative sensing and data transmission in 6G-IoT contexts. Because of their quick response time, remote operating capability, and controllable agility, UAVs are well suited for emergencies, such as anti-terror battlefields or catastrophe areas following earthquakes and floods [122]. When terrestrial infrastructure is rendered inoperable, UAVs can be launched to quickly restore wireless network coverage and communicate with the outside world [123].

### 5.10. Massive Ultra-Reliable and Low-Latency Communications

Remote health monitoring and other healthcare domains require low-latency communications (below 1 ms) with a reliability requirement of greater than 99.99% to achieve nearly real-time health provision and fast and reliable remote diagnosis. URLLC is predicted to be achieved in the 6G era, which will allow for the support of future IoT services through low-latency and dependable communication [78]. For example, the mURLLC is expected to support the timely and highly reliable delivery of massive health data to facilitate remote healthcare, with the goal of providing better medical services to patients in remote areas while also reducing regional imbalances in the health workforce. Smart factories can use mURLLC to automate mission-critical activities such as automatic production and remote robotic control. Its outstanding characteristics make mURLLC well-suited for supporting transportation systems by enabling high-reliability and timely data sharing among vehicles, infrastructure, and pedestrians. This contributes to improved smart ambulance systems and high efficiency in vehicular networks. Because of this, mURLLC can provide an extensive range of mission-critical services, including fast fault diagnosis and accurate location, dependable fault isolation, system restoration, and remote decoupling protections.

This section is summarized in Table 4, and an illustration of these technologies can be seen in Figure 8. The figure block lists the enabling technologies that will drive the evolution of 6G systems and play a significant role in enhancing m-health’s QoE as discussed in this section. The 6G will integrate the physical, biological and digital world together, as shown in this figure’s pictorial representation.

## 6. Applications Depending Heavily on 6G

A summary of the applications is given in Table 5 and Figure 9. The figure summarizes the key m-health applications that belong to 6G and will become a reality with the deployment of the 6G communication systems. The table is intended to serve as a guide for the applications that will be discussed in this section. The brief descriptions are highlighted in order to provide the reader with a foundational understanding of the applications, which will be further developed as the section unfolds.

### 6.1. Holographic Communications, Augmented and Virtual Reality

Overall, 6G will create new opportunities for many m-health applications, such as providing 3D services using holographic communication, which will revolutionize smart healthcare systems, especially telesurgery. Doctors will be able to diagnose patients remotely using holographic communication, which will reduce the financial and physical burden placed upon them by the medical system [107]. For improved diagnosis, augmented reality and holographic communication can be integrated. The primary goal of VR/AR in healthcare is to enable the transmission of high-resolution multimedia (video, audio, and voice) [124,125]. One of the drawbacks of 5G-enabled VR/AR in healthcare for multimedia transmission is the lack of a real-time platform for end-users [55]. AR enables viewing of the inside of a patient’s body, where doctors can adjust the depth of a specific body area and can be expanded to improve visibility. It will be extremely beneficial while doing difficult procedures/surgeries that involve a significant degree of risk.

Overall, 6G will enable the creation of a smooth and high-resolution display for remote medical education or diagnostics [126]. Combining AI and 6G approaches also enables seamless 360-degree video streaming at edge nodes with high packet delivery, seamless connectivity, interoperability, and convergence. The 6G network has traditionally favored highly interactive and unique video-coding techniques that enable the distribution of 360-degree on-demand and live video at a quicker rate via edge devices, which is critical for VR/AR-enabled healthcare applications. Due to the rapid adoption of mobile edge computing and edge caching technologies and trends, 6G ensures that latency and load at the backhaul are distributed equally across end-users, whereas 5G does not [127]. To deliver these videos or holograms, current 5G KPIs such as data rate, bandwidth, and dependability become insignificant [128]. These services must be able to reproduce real experiences to the user; hence, QoE is of high importance. Overall, 6G guarantees high QoE, proving itself to be the best technology for hologram/AR/VR applications [129].

### 6.2. Five Sense Communications and Tactile Internet

It is anticipated that 6G will be capable of supporting a 5D communication system with five senses, namely, sight, hearing, smell, touch, and taste [130]. Additionally, 6G demands an exceedingly high data throughput while also having incredibly low latency. Furthermore, the sensors must be capable of reproducing the five senses from faraway locations to give users a realistic experience, and as a result, 6G is primarily concerned with the QoE [108]. Another feature that is projected to be available on 6G networks is the tactile internet, which will enhance the user experience by including factors such as weight, temperature, and speed [131]. Tactile internetworking will also combine a sense of touch or taste with audio, video, or other responses. Such considerations are critical in various situations when the need for prompt remote control is required. Remote surgery, manufacturing, and navigation are examples of such applications. It is a good example of service enhancements that encourage the usage of high-precision networking [9].

The 6G networks will allow real-time data transfer, control, and feedback via touch. Using this method, one can simultaneously convey touch, sensations (sense), and information, allowing for a more interactive experience. In a smart healthcare system where an injured patient may only convey emotions through visualization, a smart headband can reconstruct brain impulses and display them as a 3D video of the patient’s imagination, which the caregiver may see in real-time using mobile networks. Furthermore, a group of people who do not speak the same language can communicate using their imaginations, and disabled people can utilize haptic communication to open doors or control gadgets. These haptic communication methods will allow them to express information through touch. It is one of the expected use cases for 6G networks, in which the network can sustain substantially higher data rates than 5G can.

QoE has been extensively investigated and considered an important aspect of multimedia (engaging more than three human senses) [132]. Tactile internet communication may demand strict latency requirements, reliability, and user experience, which can be provided by 6G mURLLC to render an immersive, synchronized effect to the human senses. The 6G-driven communication architecture, consisting of a Radio Access Network and a Core Network, satisfies the fundamental requirements for realizing the 6G-enabled Tactile Internet [29]. The following are the essential tasks of the 6G RAN in the Tactile Internet ecosystem to achieve this goal: (i) radio access technologies (RATs) such as mmWave, massive MIMO, and full-duplex; (ii) tactile QoE/QoS for tactile applications (iii) efficient packet delivery via reliable radio protocols and the physical layer; and (iv) novel medium access control (MAC) techniques that optimally control air-interface conflicts. The following are the primary functions of the 6G CN linked with the tactile internet: (i) dynamic provisioning of QoS; (ii) edge-cloud access; and (iii) security.

### 6.3. Wireless Brain-Computer Interactions (BCI)

Brain–computer interactions are another form of haptic communication in which humans interact with their environment via haptics and control it using digital gadgets such as brain-embedded wireless chips that respond to human emotions [35,133,134]. Traditionally, BCI applications were limited to those healthcare scenarios in which humans use brain implants to operate prosthetic limbs or nearby computing devices. The recent applications of wireless BCI and implants will revolutionize this industry and provide new use-case scenarios that necessitate 6G connectivity. Individuals will interact with their surroundings and other people utilizing discrete devices, some worn, some implanted, and others embedded in the world around them, employing wireless BCI technologies instead of cellphones.

The nervous system processes the data from the sensor system, which comprises receptors, neurons, etc. The format for comprehending is based on 6G wireless technology with a form of wireless biosignal processor that receives brain signals from computer equipment [135]. Following that, it analyzes, controls, and transforms these signals into electronic instructions that will be processed to determine the level of attention, locate the attention, and calculate the time required to complete the desired task, among others [35,41]. The BCI is a process for controlling the items we use daily in our smart home to transform into a smart society, and these things are all related to the m-health systems [136]. Compared to 5G, wireless BCI services necessitate fundamentally different performance metrics, including guaranteed Quality of Physical Experience (QoPE), which incorporates QoS, QoE, and human perceptions. Wireless BCI services also require high speeds, ultra-low latency, and great reliability, making 6G the first to enable a new class of applications that leverage several human cognitive senses. When considering future spectrum demands, BCI is expected to be one of many applications supported by future wireless networks [137].

### 6.4. Intelligent Ambulance Services

Overall, 6G is intended to produce self-driven, steering-less, completely AI-driven automobiles [138]. As a result of 6G’s capacity to communicate in real-time, the automobiles will monitor passenger health, such as blood pressure, heart rate, and temperature. In an emergency, intelligent automobiles will transform into intelligent ambulances. It is achieved by remote doctors employing mix-virtual reality/holograph communication over 6G. In smart ambulances, real-time medical imaging data transfer demands high data rates, which field hospitals often lack due to restricted or nonexistent network connectivity.

Communication between intelligent ambulances with medical equipment and hospital workers requires exceptionally low latency [139]. Autonomous ambulances and teleoperated vehicles can also benefit from edge intelligence supporting wide coverage to improve mobility [140]. A relatively recent concept called joint radar communication (JRC) can be used to enable reliable sensing, and vehicle-to-everything (V2X) communication in smart ambulances [141]. Furthermore, REM- and AI-assisted cognitive radios can solve handover concerns (delay, jitter) due to patient mobility with implanted medical devices. Additionally, 6G-based URLLC can be used to enable connected ambulances in future healthcare by allowing physicians and paramedical staff from the hospital to broadcast real-time video with high color resolution for reliable diagnosis at reasonably fast speeds (up to 100 km/h) [142]. For example, EEG and ECG data from clinical examinations can be collected aboard, and hospital doctors can deliver urgent indications to paramedical staff in the ambulance via URLLC-based real-time teleconsultation. In this case, an extremely short survival time (less than two milliseconds) must be guaranteed, despite the ambulance’s fast speed. By leveraging the benefits of 6G networks and AI, the traffic control system can investigate and assess the road traffic load, topology, and accidents in real time, facilitating the intelligent ambulance services [143].

### 6.5. Remote Health Monitoring of Patients

Transmission of patient vital signs is an important part of m-health applications, and the amount and frequency of information that is transmitted are dependent on the patient’s needs. The 5G services are inadequate to meet the evolving healthcare technology, and remote healthcare privacy and security [139]. The 6G integration will revolutionize m-health technologies such as mURLLC, and THz communications could support ultra-low-latency healthcare data transmission and speed up medical network connections between wearables and remote doctors [144]. Healthcare domains such as remote health monitoring necessitate low-latency communications (below 1 ms) with a reliability requirement of greater than 99.999 percent to achieve near-real-time health provision with a fast and reliable remote diagnosis.

An aging population is driving the development of novel healthcare provisions that move away from the traditional hospital-based system. These patients can be observed in their homes remotely using 6G technology by identifying critical events, such as the freezing of one’s gait and wandering behavior. The IW would make it possible to monitor patients hidden behind large objects [116]. Furthermore, the placement of IW at specific locations throughout the house would direct the beams so that signals could reach the patient-monitoring equipment. In addition to generating multiple beams [145], the IW will be useful for monitoring multiple patients in a nursing home. The monitoring devices would typically relay back patient data to a hospital network server with the assistance of the IW.

Continuous health monitoring is critical in healthcare, especially for patients with sudden diseases (e.g., cardiovascular diseases). In this system, the sensors transmit the measurements (based on patient-specific thresholds, timing and frequency as specified by a healthcare provider) via a signal processing module that further transmits to different network destinations such as hospital servers, emergency stations, remote clinicians, and others. The 6G-based healthcare systems will pre-upload data-driven models that can detect abnormal health statuses. Noninvasive sensors can continuously collect patient physiological data and send it to the cloud. The cloud models will analyze the data and send an alert if sudden diseases occur. The 6G technology will be critical for real-time monitoring and disease notification. It will also spur the development of hospital-to-home (H2H) services to replace ambulances [107].

6G networks will allow for large amounts of data to be transmitted at a faster rate, allowing for network-assisted consultations with specialists. Overall, 6G will encourage the development of these smart medical services [146] and provide a strong enough network to respond on time [147]. Health-monitoring technology has the potential to reduce the need for patients to see their doctors in person [14]. Instead, the practitioner could conduct a teleconsultation via video conferencing, with the necessary medical data shared via remote monitoring systems.

### 6.6. Telesurgery

The advent of 6G networks will be a game-changer because it will reduce space and time constraints [148]. Critical applications such as remote surgeries are extremely sensitive, requiring a latency of less than one millisecond (nearly 0.1 ms) [149]. Remote robotic surgeries will necessitate extremely high data precision and reliability and a high data rate for data exchange and control signals between two remote healthcare facilities via the mobile network. Overall, 6G will be the most suitable candidate to meet these requirements because its vision is to provide at least 99.99999% data reliability and a 1 Tbps data rate. The latency requirements are the primary concern in the case of remote surgeries. Unlike previous generations of networks, the 6G should aim for both minimum and maximum latency requirements. During remote surgery, some aspects of the transmitted data must arrive at the destination within a specified maximum latency, while others must arrive within a specified minimum latency. The combination of these minimum and maximum latencies will result in coherent data at the receiver, which is a critical requirement for remote surgery [148]. Recently, a telesurgery system based on UAVs and blockchain was investigated in [150] in the context of 6G. The blockchain is integrated into the robotic system because of the security risks in existing mobile surgery networks. Each robot acts as a data node, allowing surgical data to be securely stored in the database ledger without needing centralized authority. Smart contracts, in particular, are being used to provide automatic authentication for health data requests and control over health data sharing during surgery [103].

Meanwhile, using the THz bands in 6G-based healthcare networks, the mURLLC technology is ideal for meeting future requirements in terms of ultra-high data rates of medical data communications [151]. mURLLC is especially important in hospital-based telemedicine, where doctors can remotely monitor and manage surgery procedures using real-time video streaming from medical robotics and assistant devices connected by 6G core networks [28]. Computer-assisted operating room simulations in VR are highly beneficial for both educational and clinical settings [14]. Using multimedia components to create real-time surgical procedure simulations in a controlled and realistic setting (i.e., a virtual theater) allows the operating surgeon to perform preoperative planning and practice, surgical education, and training. Where the surgeon’s technical skills often determine the surgical outcomes, such systems have proven useful. In particular, VR/AR for surgical planning and education has the potential for medical home applications (i.e., virtual simulations aimed at increasing the emotional comfort of older and disabled patients at home).

### 6.7. Wireless Body Area Networks and Molecular Communication

Human Bond Communication (HBC) is the ability to integrate all five sensory features of humans. In humans, information is exchanged through biochemical and physical processes that involve the synthesis, transformation, emission, propagation, and reception of molecules. This information exchange is referred to as molecular communication in telecommunication engineering [152]. With the advent of NanoThings, molecular communication has emerged as one of the most important research areas for revolutionizing Medical-IoT [153].

Advanced nanotechnology has the potential to enable the manufacture of nanodevices such as nanorobots, implantable chips, and biosensors with critical applications in fields such as nanoscale sensing and biomedicine [154,155]. The application of nanotechnology in biomedicine, in particular, has generated interest due to its potential to perform tasks such as intelligent drug distribution through blood vessels and monitoring of body organs, both of which can significantly improve human healthcare outcomes. Connecting nanodevices to the internet or forming networks (i.e., the Internet of Nano-Things) enables effective communication and information transmission. In biomedicine, the Internet of Bio-Nano-Things (IoBNT) enables the connection of nanodevices and biological entities. In the IoT, molecular communication enables the IoBNT by allowing biological molecules to communicate and transport information between nanodevices [154]. Additionally, when the IoBNT is combined with body area networks, which are short-range wireless networks comprised of wearable monitoring devices/sensors and sensing devices embedded in or on the body, comprehensive solutions for healthcare enhancements can be provided.

## 7. Challenges

Several challenges are associated with m-health multimedia data.Despite recent efforts to address QoE control and management difficulties and the rapid evolution of cellular communication systems, there are still significant gaps in certain areas that require extensive research and investigation [45]. To address the provisioning of QoE to end users, additional research on multimedia m-health services should be conducted in the following critical areas that have received little attention in the past:

### 7.1. Spectrum Management for the m-Health Multimedia Applications

In future wireless networks, meeting the demands of formidable and diversified 6G use cases involves high resource efficiency and more flexibility and scalability [156]. Resource expansion is critical as the networks are expected to go beyond their statistically assigned resources and take advantage of available resources based on spatial and temporal constraints. With new services and applications such as 3D holographic image and presence, 5D communications, smart clothes and wearables, and completely autonomous vehicles, 6G will take resource challenges to the next level [41]. Furthermore, as predicted by ITU-R, connectivity would reach an extreme level by 2030 [157], with nearly 120 billion subscriptions, including M2M subscriptions, and exponentially growing traffic [158].

Managing 6G spectrum resources and interference issues demands updated spectrum management technology and appropriate sharing policies [159]. Effective management is critical for optimal resource utilization with high QoS and QoE. The 6G radio must be agile and adaptive beyond the 5G NR definition. It must be capable of utilizing multiple channel bandwidths across a wide spectrum, supporting cell-free communications, implementing ultra-massive MIMO, and operating on a discontinuous spectrum [35]. The requisite flexibility can also be achieved by integrating network softwarization and network slicing more deeply into core and transport networks. Additionally, the stringent operational settings, ranging from ultra-reliable and low-latency scenarios to massive connectivity, necessitate a considerably more flexible radio capable of spectrum sharing [36]. In that context, capacity expansion via adaptive strategies such as spectrum sharing is critical in serving applications envisioned by the 6G ecosystem [160]. There are many survey articles about CRNs in the literature; however, most of them overlook the rigorous requirements of multimedia communications. Those considering multimedia transmission over CRNs failed to address the difficulties of QoS/QoE provisioning [161].

Building a viable solution for dynamic spectrum allocation is a major issue in CRNs. A simplified QoS uniform assumption for spectrum assignment must be explored to solve this problem. The explicit QoS needs of Secondary Users (SUs) for various m-health multimedia services must also be considered; otherwise, SUs will continually hand off to other channels in search of the best available channel, resulting in quality degradation. As a result, providing QoS and QoE for multimedia transmission over CRNs poses significant issues.

### 7.2. Standardizing QoE Mapping to Network Performance Indicators

Despite the numerous research initiatives and remarkable progress made in recent years, m-health multimedia communication still has a number of unresolved concerns that need to be addressed. In the m-health services and IoT applications, user satisfaction with multimedia content is measured using subjective QoE indicators. In recent studies on QoE evaluation for multimedia applications, packet loss, MOS, DMOS, and user satisfaction are the key QoE indicators used. For M-IoT evaluation, other QoE indicators such as buffering time, data rate, interruption time, and failure rate should be employed [162]. Jitter, end-to-end delay, bit error rate, and frame loss are the most commonly used network performance measures (QoS metrics) in M-IoT research. To improve multimedia quality and network services, both QoS and QoE measurements should be examined together [163].

QoE management in 6G-enabled m-health will be a difficult task, as QoE is expected to be maintained competently and autonomously for each patient and the accompanying needed service. As a result, a data-driven architecture for individualized QoE management in the next-generation healthcare network has been suggested. The suggested architecture must meet some requirements to capture the user’s subjective characteristics. A definable cognitive computing capability must be built into the architecture, and the design should be based on a data-driven model that can predict customer demand. To maintain a satisfying QoE, the architecture must efficiently manage communication resources based on the QoS requirements, and the expected user-centric service [164].

### 7.3. The Role of Contextual and Device Information in QoE

Context awareness was initially studied in the context of pervasive computing, in which relevant information about the user and its environment, such as location, user identification, etc., was obtained and used by the computing system to adapt accordingly. Numerous context-aware approaches for optimizing cellular network operations, including radio resource management, mobility management, and congestion control, have been developed. Current RAN architectures are often optimized for a single RAT, which frequently under-utilizes available heterogeneous resources. Additionally, such an architecture cannot ensure consistent QoE for users traveling between various RANs. Context awareness becomes critical in these situations, allowing for more precise and intelligent administration. The 6G network must be planned to be entirely context-aware. The network must constantly be aware of all devices in its environment and their capabilities. Various open difficulties are associated with achieving total context awareness and administering such a system efficiently. Several critical issues in this area include: managing conflicting changes in a system caused by multiple applications or network functions based on the same context, monitoring and collecting context information without overwhelming the core network’s operation, and maintaining the acquired context’s recentness by aligning the temporal scales of monitored information [44]. The context-aware quantized QoE model improves user experience while ensuring resource efficiency, while the invalid action masking approach reduces invalid RATs to accelerate convergence [164].

In recent years, video quality assessment studies have focused on incorporating the context awareness of the application in order to assess video quality. Context awareness includes considering factors such as the video content, video characteristics such as frame resolution, frame rate, etc., the delivery mechanism, the application in which it is used, the end user’s requirements, the devices on which the videos are streamed, etc. In medical video streaming, it is particularly important to consider the context of the system. This is because the quality of the video in m-health applications should not just be of good visual quality, but also retain its quality so that the video is suitable for diagnosis. Here, context can be defined as any information that aids in determining a user’s QoE. There are two types of context: static and dynamic [64]. Dynamic context evolves over time and is difficult to predict, but static context does not change frequently. The user’s application choices, security needs, and cost are all examples of the static context. The context in real-world settings can be highly dynamic and stochastic, i.e., it can change rapidly and is uncertain and imperfect; have a variety of temporal characteristics; have several alternative representations, interconnected or distributed; and it may not be available at any given time. Context gathering and processing should be done as soon as possible because it can lose its correctness. User position, velocity, network load, battery power, memory/CPU utilization, presence, and signal-to-noise ratio are all examples of dynamic context. QoE can be considered as a function of contextual factors (represented as a vector).

Measurement and prediction of QoE might include a vast parameter space with multiple context attributes and QoE parameters. Different context attributes may influence different QoE parameters. The QoE parameters can also interact with one another. As a result, choosing important context traits and QoE parameters and establishing links between them can be difficult. These connections are frequently non-linear and difficult to quantify. This needs the creation of unique QoE modeling methodologies that can effectively model various context features and QoE parameters. The new QoE models should be more than just conceptual; they should also address the issues of QoE measurement and prediction.

### 7.4. Influence Factors (IF)

In general, the m-QoE perceived in m-health systems is influenced by the system, the context and human Influence Factors (IFs), and these need to be taken into account during the design stage [9,165]. The first refers to what can be called the user’s demographic composite. These factors include an individual’s age, gender, culture, personality, mood, and interests. Clearly, these are not static effects and are subject to change throughout time and space. Context modifies human IFs by incorporating information about the user’s location, surroundings, time, activity, co-occupants of shared space, technical affluence, and economic capabilities. Finally, system IFs are concerned with the content, the configuration of the media, the network condition, and the device (or gear). A detailed understanding of the IFs and their impact on the QoE of a specific service is a key requirement for successful QoE management with the overall goal of optimizing end-user QoE while utilizing network resources efficiently and retaining a contented user base. It is required to identify and comprehend many influence factors (IFs) and perceptual dimensions from the perspective of various players in the service delivery chain in order to ascertain their impact on QoE. This is a necessary precondition for QoE management since it establishes the parameters to be monitored, assessed and ultimately utilized to develop, test, and apply QoE control and optimization techniques.

Because QoE is a relatively new term, content providers, service and network providers, as well as academics are confronted with new issues in delivering, evaluating, and controlling QoE. Then, a preliminary step would be to investigate and analyze the QoE-influencing parameters (IFs) [25]. Due to the subjective nature of the QoE, it is difficult to predict. Thus, prior to evaluating the overall service quality, it is necessary to ascertain the aspects that influence users’ perceptions [26].

### 7.5. Mobility Management

Mobility management in 6G will be complicated as the networks will be extremely heterogeneous, multi-layer, large-dimensional, and dynamic [80]. To provide a positive user experience, optimize resource utilization, and reduce network signaling overhead, intelligent mobility management is critical. It can be accomplished by learning and predicting user movement and optimizing handover settings. Numerous academics have attempted to understand and predict user mobility behavior using machine learning approaches. For example, in [28], the back-propagation NN is used to forecast the next cell a user will move towards by learning the user’s mobility model and then predicting the next cell. Additionally, the optimization of handover (HO) parameters is a critical problem for 6G networks, as it can affect various factors, including coverage capacity, energy usage, interference, and user experience. As a result, various works utilizing machine learning have been devoted to the subject.

The 6G networks plan to support high speeds to counter mobility, but this will necessitate regular handovers. Additional service needs, such as high data flow, high dependability, and low latency, may complicate efficient changeover [44]. When a mobile device requests video content, it is often issued from servers and subsequently traverses the mobile carrier’s CN and RAN. Many concurrent streams would create enormous demand on the backhaul side. Additionally, the unpredictable characteristics of the wireless channel (e.g., fading, multi-user interface, peak traffic loads, etc.) may pose a challenge for monitoring user QoE by imposing additional stress on the cellular network. Nonetheless, distributing streaming content is a challenge, as the link between servers hosting the needed information and consumers can introduce delays in data transfer, impairing the user experience. By bringing content closer to the consumer via caching, numerous hurdles such as network load and latency can be addressed, resulting in an improved QoE [45].

Handover decisions should be made carefully in these HetNets environments to guarantee that video applications meet their criteria for low packet loss and delay, as well as QoE support [166]. For example, a user traveling in a vehicle may pass through multiple geographical areas in a short period of time, necessitating repeated handovers in order to retain connectivity with the video service provider [167]. Thus, independent of the network circumstances and characteristics or the user’s mobility, the handover must occur without compromising the QoE, i.e., a video devoid of ghost effects, pixelization, or screen freezing.

Handover methods in HetNets use radio (e.g., signal strength) or QoS context factors as a key metric for handover decisions. However, considering it alone or in combination with QoS is inefficient for mobile multimedia. Determining, controlling, and improving the quality of multimedia applications is not limited to QoS techniques. Handovers in heterogeneous networks should address subjective characteristics of video content, such as user perception and pleasure. QoE must be quantified and integrated into decision-making to improve handover decisions while meeting mobile user video service requirements. The current HetNets must adopt QoE-centric methodologies. Cognitive radio (CR) is a prospective candidate technology for enabling HetNets. HetNets based on CR can give better-suited QoE for multimedia applications that require a lot more bandwidth [161].

## 8. Conclusions and Future Works

The vision for the 6G network indicates that it will vastly surpass the 5G network in terms of meeting the high requirements of future use cases such as telehealthcare, augmented/virtual reality, and haptic internet. With customer demand for video services increasing, QoE is critical for service providers and network operators. This paper aims to draw attention to the complex elements of QoE in 6G networks and the limitations of current definitions in the context of m-health applications. We identified some interesting technologies in 6G networks that will significantly impact the healthcare business and improve the QoE for m-health applications. We provide an in-depth description of artificial intelligence, intelligent reflecting surfaces, THz communications, blockchain, and extended radio, among other technologies, and how they will revolutionize m-health. We covered numerous healthcare applications that require 6G networks to deliver extreme data speeds, ultra-low latency, extreme dependability and availability, vast scalability, extreme power efficiency, and extreme mobility. In the end, we explore the difficulties that 6G will encounter while implementing new healthcare applications.

In addition to the aforementioned aspects, it is crucial to address the existing gaps in medical data transmission within the context of 6G networks. While extensive research has been conducted in this area, the absence of a standardized format universally compatible with other wireless devices poses a significant challenge. Future telemedicine designs should prioritize the resolution of concerns pertaining to scheduling mechanisms, data security, privacy considerations, and the adoption of advanced compression techniques for medical video streaming. Furthermore, the design of wearable medical devices must carefully consider factors such as energy efficiency, radiation absorption rates from wireless sensors, and the provision of adequate skin protection, ensuring the safety and well-being of users. By addressing these critical issues, the implementation of 6G networks in healthcare applications can be further optimized to enhance the quality of experience for m-health services.

## Figures and Tables

**Figure 1 sensors-23-05882-f001:**
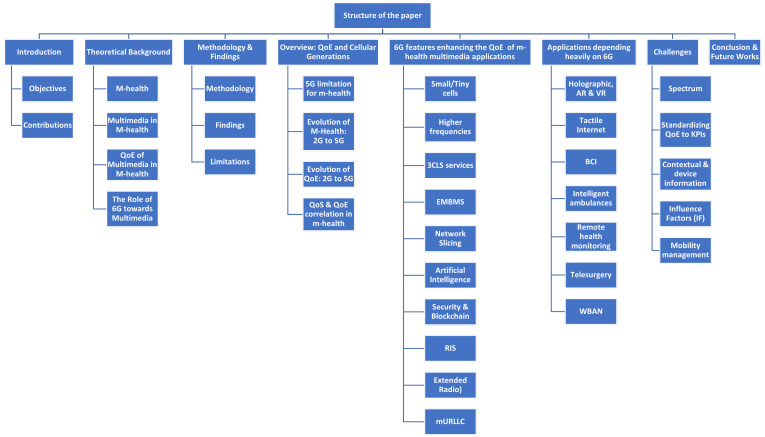
Illustration of the topics and technologies covered in the paper.

**Figure 2 sensors-23-05882-f002:**
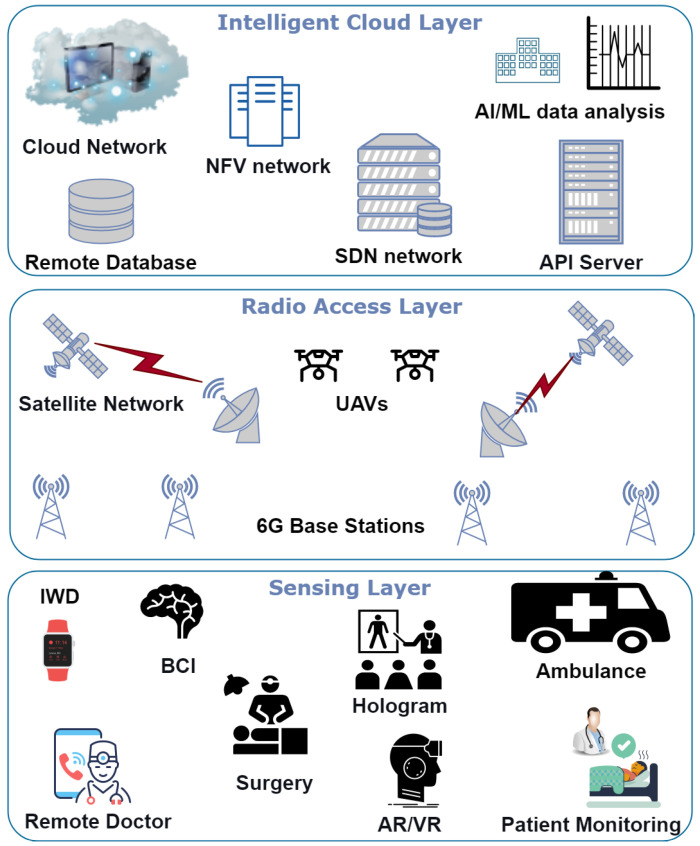
The 6G healthcare network layers.

**Figure 3 sensors-23-05882-f003:**
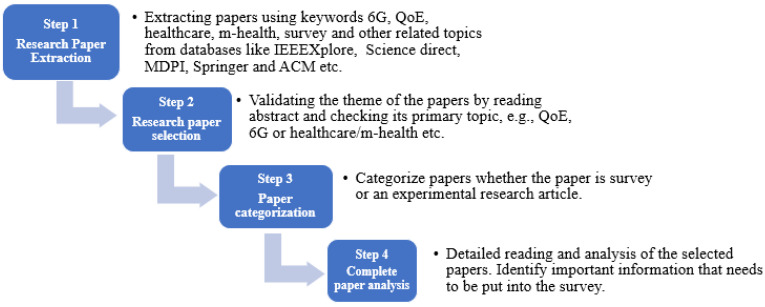
Research methodology.

**Figure 4 sensors-23-05882-f004:**
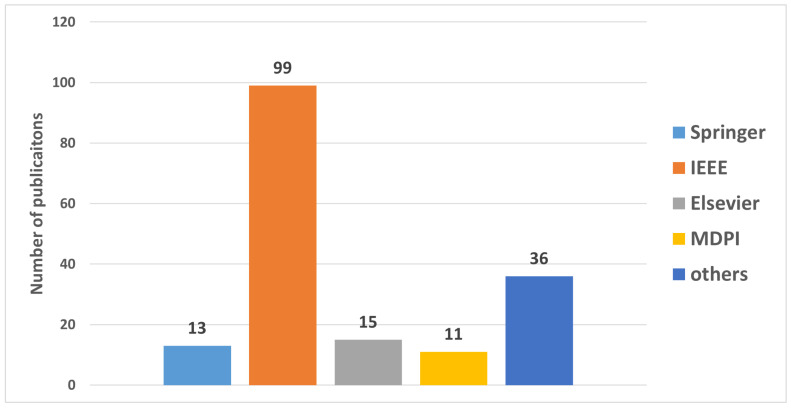
Distribution of cited papers by publishers.

**Figure 5 sensors-23-05882-f005:**
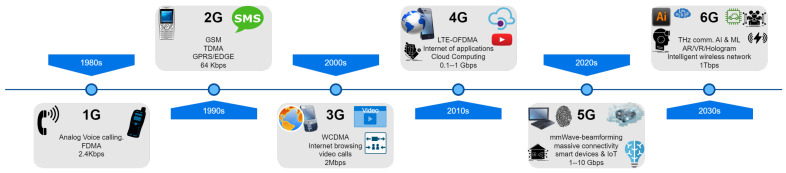
Evolution of cellular communications: from 1G to 6G.

**Figure 6 sensors-23-05882-f006:**
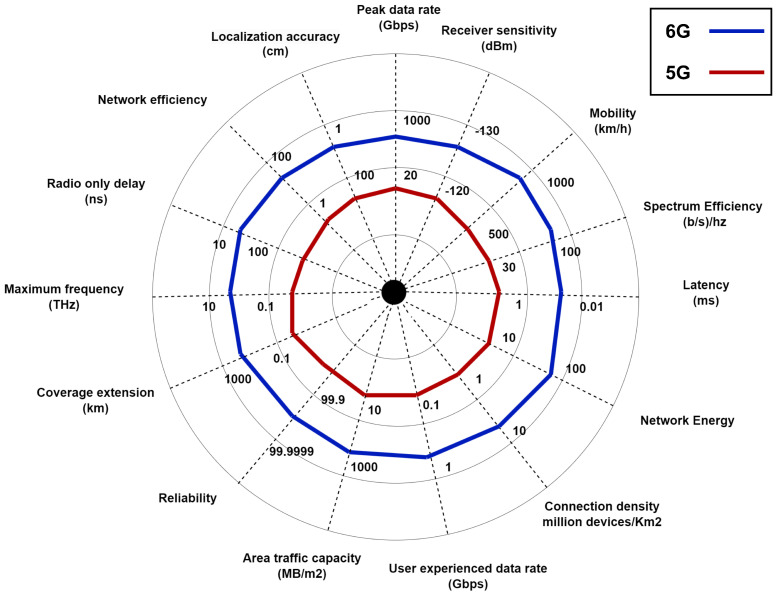
Comparison of 6G and 5G capabilities.

**Figure 7 sensors-23-05882-f007:**
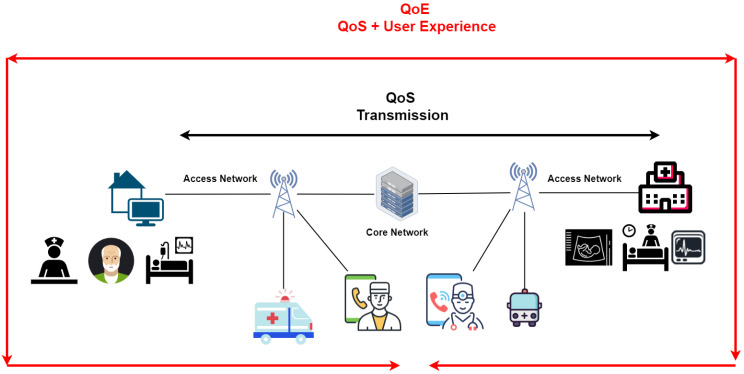
QoS and QoE in m-health applications.

**Figure 8 sensors-23-05882-f008:**
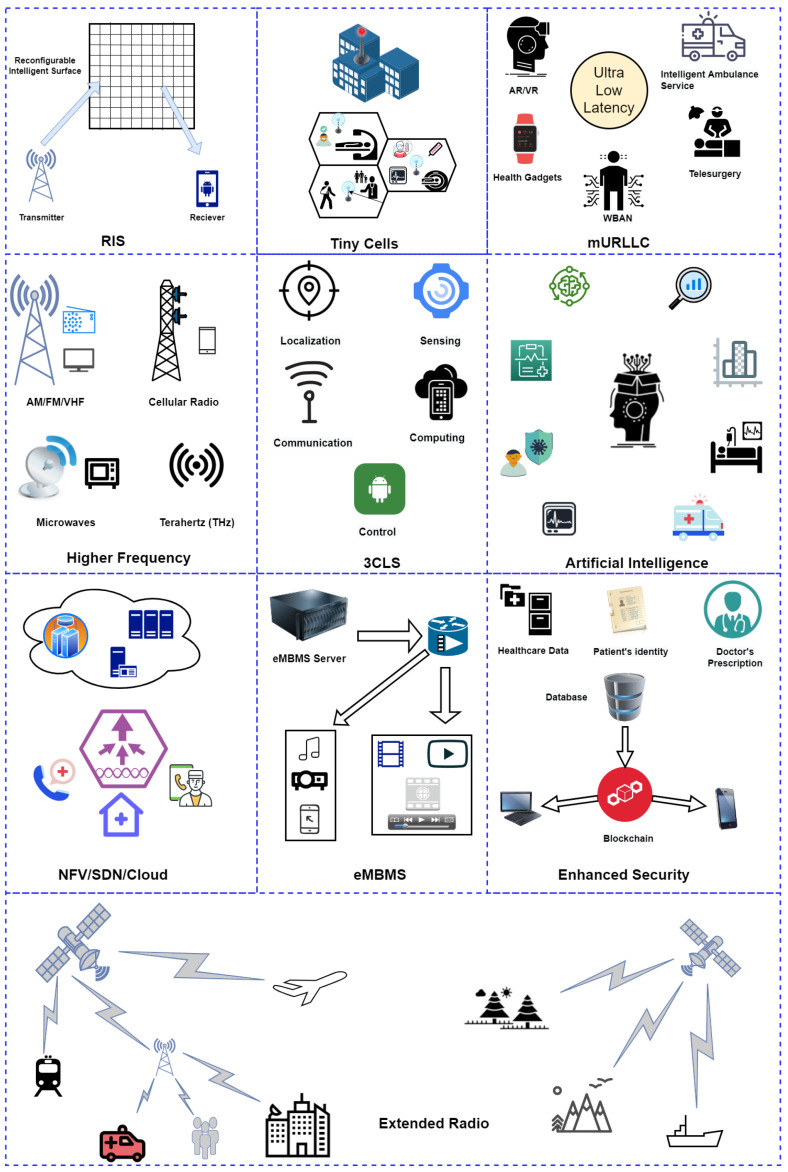
The 6G features significantly enhancing the QoE of m-health multimedia applications. The 6G features are summarized in this figure to highlight the essential components of every service and to facilitate it to the readers.

**Figure 9 sensors-23-05882-f009:**
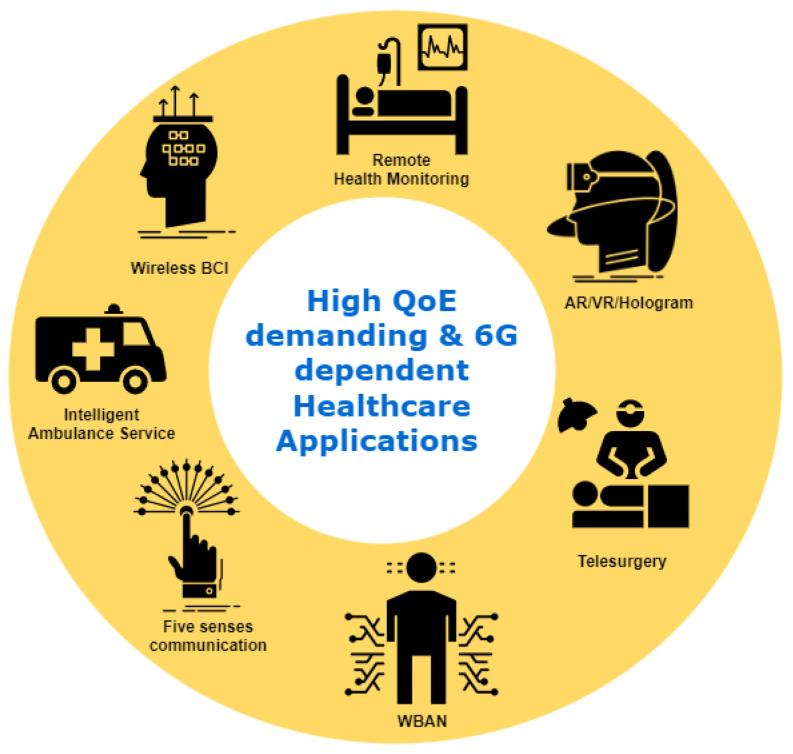
Applications depending heavily on 6G.

**Table 1 sensors-23-05882-t001:** Subjective assessment metrics for multimedia quality evaluation.

Metrics	Description
Mean Opinion Score (MOS)	A widely used metric that evaluates the overall quality of multimedia content by asking users to rate it on a scale from 1 to 5 or 1 to 10.
Differential Mean Opinion Score (DMOS)	Measures the difference between the user’s rating of the quality of two versions of multimedia content.
Single stimulus (SS) or Absolute category rating (ACR)	The most basic method of subjective assessment, where the viewer rates the quality of a stimulus on a predefined scale.
Absolute category rating-hidden reference (ACR-HR)	A variant of SS/ACR method where the reference is hidden from the viewer.
Degradation category rating (DCR) or Double stimulus impairment scale (DSIS)	The viewer is presented with two stimuli—the original and the degraded version, and the viewer is asked to rate the quality of the degraded stimulus.
Double stimulus continuous quality scale (DSCQS)	Similar to DCR/DSIS method, but the viewer is presented with a continuous scale to rate the quality of the stimulus.
Pair comparison method (PC)	The viewer is presented with two stimuli, and the viewer is asked to select the stimulus with better quality.
Subjective assessment methodology for video quality (SAMVIQ)	A standardized subjective assessment methodology for video quality evaluation that involves multiple test conditions and scales.
Single stimulus continuous quality evaluation (SSCQE)	The viewer is presented with a stimulus and asked to continuously rate the quality on a scale.
Simultaneous double stimulus for continuous evaluation (SDSCE)	The viewer is presented with two stimuli, and the viewer is asked to continuously rate the quality of both stimuli.

**Table 2 sensors-23-05882-t002:** Objective assessment metrics for multimedia quality evaluation.

Metrics	Description
Peak signal-to-noise ratio (PSNR)	Measures the difference between the original and the compressed multimedia content.
Structural similarity index (SSIM)	Measures the similarity between two images or video frames.
Multiscale-SSIM (MS-SSIM)	An extension of SSIM that considers the structural information of images at multiple scales.
Moving picture quality measure (MPQM)	Measures the quality of the compressed video based on the perceptual quality of individual frames.
Video quality metric (VQM)	Measures the quality of the compressed video based on the distortion of individual frames.
Visual information fidelity (VIF)	Measures the visual similarity between two images.
Visual signal-to-noise ratio (VSNR)	Measures the difference between the original and the compressed multimedia content based on the perceptual quality of individual frames.
Motion-based video integrity evaluation (MOVIE)	Measures the quality of the compressed video based on the amount of motion.
Video multimethod assessment fusion (VMAF)	A perceptual video quality metric that combines several objective metrics to predict the subjective quality of the video.
Spatio-temporal reduced-reference entropic (STRRED)	A reduced-reference metric that measures the quality of the compressed video based on the spatiotemporal distribution of entropy.
STRREDopt	It is a computationally efficient variant of STRRED.
Spatial efficient entropic differencing for quality assessment (SpEED-QA)	A reduced-reference metric that measures the quality of the compressed video based on the spatial distribution of entropy.
No reference bitstream (NR-P)	A no-reference metric that measures the quality of the compressed video based on the statistical properties of the bitstream.
Pixel-based no-reference (NR-B)	A no-reference metric that measures the quality of the compressed video based on the statistical properties of the pixels.
Blind/referenceless image spatial quality evaluator (BRISQUE)	A no-reference metric that measures the quality of images based on the statistical properties of the pixels.
The natural image quality evaluator (NIQE)	A no-reference metric that measures the quality of images based on natural scene statistics.
Psychovisual-based image quality evaluator (PIQE)	A no-reference metric that measures the quality of images based on the human visual system.

**Table 3 sensors-23-05882-t003:** Categorization of survey papers according to primary topic.

No. of Papers	9	44	14	4	1	2
Primary Topic	Healthcare	6G	QoE	Healthcare + 6G	Healthcare + QoE	6G + QoE

**Table 4 sensors-23-05882-t004:** Summary of the 6G features contributing to the QoE enhancement of m-health multimedia applications.

6G Feature	Description
Tiny Cells	Work well with higher mmWave and THz frequencies. They are simple to install and configure, with the ability to restrict access to a specific set of users present at home or in an office environment.
Higher Frequency	High-frequency spectrum (THz) exploration is being pursued with the goal of achieving data rates of one terabyte per second (TB/s).
3CLS	In addition to the wireless communication that earlier generations offered, this generation adds computing, control, localization, and sensing. Supports various new applications.
eMBMS	Video compression methods and new technologies targeted to video applications benefit from the eMBMS bearer. MIMO technology is used by eMBMS to provide consumers with Multicast/Broadcast media content that supports Quality of Experience (QoE), such as High-Definition Video streaming.
Network Slicing	Enables superior flexibility, scalability, and specialization capacity for the provision of novel and specialized services. It simplifies the process of updating and patching the program.
Artificial Intelligence	The use of artificial intelligence (AI) is proposed to enable self-sustaining wireless networks, providing a distributed (AI) architecture for model training in which each participating device learns the model locally and then communicates the updated model rather than the data. This addresses issues such as privacy leakage, huge overheads, etc.
Blockchain	Numerous benefits of immutability, disintermediation, audibility, and non-repudiation are provided by blockchain technology, which enables decentralized and distributed management of a cryptographically secure digital ledger. Secure spectrum, resource, mobility, and interference management capabilities.
RIS	Smart reflecting surfaces that can act as walls, roadways, doors, and entire buildings. RIS aids in maintaining a line of sight and obtaining a high-quality signal with less noise and interference from other users.
Extended Radio	6G is intended to integrate space–air–ground–sea mode, enabling wireless communication in planes, UAVs, and other applications.
mURLLC	Provide extremely low latency. Ultra-high reliability, high availability, massive connectivity, and scalability.

**Table 5 sensors-23-05882-t005:** Summary of the 6G m-health multimedia applications.

Application	Description
AR/VR and Holograph communications	Human communication via holograms (3D pictures in thin air) to enhance remote communication. Holographic communication is hampered by latency and excessive bandwidth, but 6G will overcome them. Perceptual AR/VR, URLLC and eMBB in 6G will enable perceptual aspects. A great candidate for better remote healthcare applications.
5 Sense communications and Tactile Internet	Allows human–machine and machine–machine interactions.
Wireless Brain-Computer Interactions	The brain–computer interface (BCI) facilitates communication between the brain and electronic equipment. High reliability, ultra-low latency, and a large data flow are required for this.
Intelligent Ambulance Service	Remotely control automobiles. These autos are also called semi-autonomous. Semi-autonomous vehicles require a reliable, low-latency wireless network. A method employing 6G-based URLLC to facilitate connected ambulances in future healthcare by enabling real-time video streaming with high color resolution for trustworthy diagnosis.
Remote Health Monitoring	The combination of mURLLC and THz communications provides a solution for remote health monitoring that allows for exceptionally low-latency data transmission and network connections between wearables and distant clinicians.
Telesurgery	A more comprehensive approach incorporating unmanned aerial vehicles (UAVs) and blockchain integration, which may provide both rapid and secure features for surgical processes, is being developed.
WBAN/IoBNT and Molecular Communications	A network of biological nanoparticles (nanomachines). Mostly used in healthcare. The 6G technology is proposed to meet loBNT’s perceptual and latency criteria. It uses mURLLC to connect nano-devices, implants, and on-body sensors to edge/cloud servers.

## Data Availability

Not applicable.

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
