# Peer review of "Exploring the Role of 6G Technology in Enhancing Quality of Experience for m-Health Multimedia Applications: A Comprehensive Survey"

_sensors, 2023, doi:10.3390/s23135882_

Round 1

Reviewer 1 Report (New Reviewer)

Thank you for the opportunity to review this article about Exploring the Role of 6G Technology in Enhancing Quality of Experience for m-Health Multimedia Applications: A Comprehensive Survey. I found the article interesting, but it could use some revisions.

While the article is interesting, my main concern is that I found it excessively long and difficult to read. I recognize the authors' focus on relevant topics, but condensing the findings for improved readability would be helpful. I respectfully suggest the authors consider synthesizing the information to make it more accessible to readers.   I acknowledge that the work presented is a narrative rather than a systematic review. Nonetheless, a more organized structure would assist readers in critically assessing the content and reaching more objective conclusions.   For example, the introduction is interesting, but it reads more like an essay than a description of the current state and the reasons for undertaking the work. Therefore, the authors should concisely justify and clearly define their objectives while shortening the section.   To ensure reproducibility, it is essential to present the methodology thoroughly and objectively. Although the review may have a narrative aspect, providing a detailed description of the procedure helps place the results in context and allows for the evaluation of potential biases by both authors and readers.   Some areas could be improved, such as narrative expressions that do not align with the scientific language required for an article. For instance, the following examples: «In what follows, we provide an overview of the different aspects used to facilitate the investigation of our research.», or «Finally, the contributions of the conducted research and the paper outline are provided.» Another example: «This section starts the discussion with the 5G limitation for m-health applications, followed by an evolution of m-health wireless systems and the quality of experience (beginning from 2G to 5G), and finally, showing the connection between the quality of 260 service and quality of experience.» As previously mentioned, a more thorough methodology description could assist in pinpointing these narrative expressions.   The paragraph on lines 239-247 refers to figures 2, 3, 4, and 5. However, figure 2 is presented before the paragraph. Figure 3 appears after it, table 3 after Figure 3, and Figures 4 and 5 are not included. In the following paragraph, figure 4 and Table 4 are mentioned, but they need to be shown at the end of the paragraph. Figures 4 and 5 are on the next page after another paragraph. To make the text easier to read, the authors should ensure that figures and tables are placed immediately after the paragraph where they are mentioned. The authors highlight that certain aspects have already been presented in Table 3, but Table 3 is shown later in the text. These aspects are especially important given that the text is already complex.   Table 3 is too detailed and lacks visual representation, making it difficult for readers to draw conclusions or understand the information presented. To improve readability, a summarized table could be provided as a bar chart showing the number of articles mentioning each aspect in the top row. If the authors wish to include Table 3, it should be added as supplementary material and not in the article's main body, as it may need to be clarified for readers.   Starting from Figure 3, the authors present their results. It would be beneficial to improve readability if they clearly distinguished their review's results from the discussion. A separate section dedicated to the findings would allow for a better understanding and assessment of potential biases before delving into the authors' intriguing discussion.   As a narrative and not a comprehensive review, before the authors conclude, it would be beneficial for readers to consider the limitations of this research. Some limitations are apparent, such as the non-systematic approach and selection biases. These limitations help contextualize the conclusions' external validity. Although the authors did not intend to provide a reproducible development, they should make it refutable, and objective evaluations like assessments of biases or limitations can increase the potential for refutability.

English usage seems correct.

Author Response

Dear Reviewer,

We appreciate your time and great efforts in handling the review process of our manuscript. Based on the reviewers’ valuable comments, we have improved our manuscript's contents and presentation qualities by carefully addressing the comments. As such, we enclose a point-by-point response in what follows. We appreciate the valuable time and kind help of the reviewer once again.

The paper titled “Exploring the Role of 6G Technology in Enhancing Quality of Experience for m-Health Multimedia Applications: A Comprehensive Survey” has been carefully revised. The corresponding change and refinements we have made and the significant modifications are summarized below:

  1. The content of the paper was reduced wherever redundant information was found.
  2. The structure of the paper has been revised, particularly the Introduction.
  3. The research methodology has also been explained appropriately in a section, along with the literature review findings.

We have enclosed a point-by-point response to address all comments in detail. 

We hope the revised manuscript can meet your expectations. Thank you again for your time and kind consideration of our submission.

Yours sincerely,

Dr. Moustafa M. Nasralla

Reviewer 2 Report (New Reviewer)

The paper represents a thorough study on the role of emerging technologies on QoS/QoE in m-Health multimedia applications.

The paper is an extensive literature review aimed at summarizing some important problems in m-Health  QoS/QoE.

The abstract is well-written and summarizes the problems but the authors should clearly state in the abstract that the paper is a review of the literature.

In the Introduction, the most important notions such as m-health,  Quality of Service (QoS), Quality of Experience (QoE), etc. The Introduction is excellent and many important sources are cited. I recommend to the authors to explain in more details the notions of QoS/QoE and to include references to the documents International Telecommunication Union for example the QoS regulations (ITU-T Supp. 9 of E.800 Series),  the vocabulary for performance, quality of service and quality of experience, etc.

Also, in recent years there are studies on QoS and QoE in overall telecommunication systems, addressing important problems related to determining the QoS. For example, an overall normalization approach for determining the QoS in overall telecommunication systems is described in:

S. A. Poryazov, et al., "Overall Model Normalization towards Adequate Prediction and Presentation of QoE in Overall Telecommunication Systems," 2019 14th International Conference on Advanced Technologies, Systems and Services in Telecommunications (TELSIKS), Nis, Serbia, 2019, pp. 360-363, doi: 10.1109/TELSIKS46999.2019.9002295.

The relations between QoS and QoE are also studied in the above paper.

In the paper:

Mutichiro, B.; Tran, M.-N.; Kim, Y.-H. QoS-Based Service-Time Scheduling in the IoT-Edge Cloud. Sensors 2021, 21, 5797. https://doi.org/10.3390/s21175797.

a mechanism that improves the cluster resource utilization and Quality of Service (QoS) in an edge cloud cluster in terms of service time is proposed. Containerization provides a way to improve the performance of the IoT-Edge cloud by factoring in task dependencies and heterogeneous application resource demands.

Also, in recent years an extension of the Petri Nets  - Generalized Nets – has been successfully used in telecare/telehealth to reduce the costs and improve the quality of the service provided. This approach is described in the paper:

Stefanova-Pavlova, M., et al. (2016). Generalized Nets in Medicine: An Example of Telemedicine for People with Diabetes. In: Angelov, P., Sotirov, S. (eds) Imprecision and Uncertainty in Information Representation and Processing. Studies in Fuzziness and Soft Computing, vol 332. Springer, Cham.

The GNs approach is a promising new direction with many applications and advantages.

The Conclusion is rather short and must be extended. Despite the numerous studies conducted in this field, medical data transmission using a standard format that is universally compatible with other wireless devices remains lacking. Future telemedicine designs need to address issues such as scheduling, data security and privacy, and medical video streaming employing modern compression techniques. Another issue is that the design of wearable medical devices needs to take cognizance of energy efficiency, the absorption rate of radiation from wireless sensors, and the need for skin protection.

The paper is of high quality. I recommend that the paper be published once the authors address adequately my critical remarks.

English is fine. 

Author Response

Dear Reviewer,

We appreciate your time and great efforts in handling the review process of our manuscript. Based on the reviewers’ valuable comments, we have improved our manuscript's contents and presentation qualities by carefully addressing the comments. 

The corresponding change and refinements we have made and the significant modifications are summarized below:

  • Appropriate citations have been included in the manuscript, as suggested.
  • The conclusion has been extended.

We have enclosed a point-by-point response to address all comments in detail. 

We hope the revised manuscript can meet your expectations.

Thank you again for your time and kind consideration of our submission.

Yours sincerely,

Dr. Moustafa M. Nasralla

Round 2

Reviewer 2 Report (New Reviewer)

I would like to thank the authors for taking into account my remarks. The paper has been sufficiently improved.

This manuscript is a resubmission of an earlier submission. The following is a list of the peer review reports and author responses from that submission.

Round 1

Reviewer 1 Report

- Is the review clear, comprehensive and of relevance to the field? Is a gap in knowledge identified?

The article has a relevant and significant character to the Health and Technology area, contemplating characteristics that can be better highlighted in a work that considers its continuity in a case study or exploratory analysis.

- Was a similar review published recently and, if yes, is this current review still relevant and of interest to the scientific community?

The article had the intention of demonstrating and building the gap that precedes the theme, which was well constructed. Although, there is still the need for scientific improvement, in face of more recent publications and that can establish a temporal linearity between the central themes and future needs.

- Are the cited references mostly recent publications (within the last 5 years) and relevant? Are any relevant citations omitted? Does it include an excessive number of self-citations?

The references and citations are up-to-date and include a recent methodological and scientific framework, favoring articles and journals with significant impact.

- Are the figures/tables/images/schemes appropriate? Do they properly show the data? Are they easy to interpret and understand?

Figures could be improved or optimized to make them clearer. The use of more professional or scientific infographics is also recommended. As a suggestion, some websites and applications can provide the illustration and representation of infographics in a dynamic way, which could enrich the work.

Reviewer 2 Report

The paper  aims to provide the first thorough survey on a promising research subject that exists at the intersection of two well-established domains, i.e., QoE and m-health, and is driven by the continuing efforts to define 6G.

The abstract of the paper defines well the problem and the contributions of the present research but must be reduced in size. Now it is more like an introduction to the problem.

The Introduction is very well written and presents all basic notions such as healthcare, m-health,  QoS, QoE, the use of  telecommunication and network technologies in the healthcare (telemedicine) domain.  The references to the problem and the terms are well-chosen. To some directions of research however no references are made.  In particular the paragraph on lines 39-53. There are recent works in a new direction of healthcare (and telemedicine) which uses the apparatus of the Generalized nets, an extension of Petri nets, which have been proven to reduce the cost  of the service and to increase the QoS (and QoE). This direction of research is described in the papers:

Stefanova-Pavlova, M. et al. (2017). Modeling Telehealth Services with Generalized Nets. In: Sgurev, V., Yager, R., Kacprzyk, J., Atanassov, K. (eds) Recent Contributions in Intelligent Systems. Studies in Computational Intelligence, vol 657. Springer, Cham.

Subsection 1.1 overviews the notion of m-health. It is well-structured and all aspects of m-health related to mobile apps are mentioned. In particular, m-health applications go hand-inhand with wearable  sensors and Internet of Things (IoT) devices. All references are on topic and represent significant contributions in the field.

Subsection 1.2 is devoted to the multimedia communications in m-health. The importance of multimedia communications to maintaining a certain level of QoS and, therefore, of QoE is also stated. However, a reference to this problem must be made and I suggest that authors make a reference in the paragraph on lines 107-114 to the following important work on enhancing the QoS/QoE in multimedia networks:

C. Mancas and M. Mocanu, "Enhancing QoS/QoE in multimedia networks," 2013 IEEE International Conference on Communications Workshops (ICC), Budapest, Hungary, 2013, pp. 637-641, doi: 10.1109/ICCW.2013.6649311.

Subsection 1.3 is devoted to the very important and extremely complicated problem of  measuring and maintaining the QoE in multimedia communications. Lines 119-127 address the QoS/QoE  estimation is service systems in general. The authors should include references to the documents of the International Telecommunication Union for example the QoS regulations (ITU-T Supp. 9 of E.800 Series), QoE requirements for real-time multimedia services over 5G networks (https://www.itu.int/pub/T-TUT-QOS-2022-1); ITU-T Recommendation ITU-T P.10/G.100  (11/2017). Vocabulary for performance, quality of service and quality of experience, etc.

where the importance of the normalization of the concepts, normalization of the indicators’ scales, etc. are discussed.  Also, there is a paper of Reichl from 2010 which explains the logarithmic nature of the QoE and the Weber-Fechner Law. Such more recent studies on the relation between QoS and QoE should be included.

Subsection 1.4  describes the Role of 6G towards Multimedia Communications. Here the references are on topic. I have no  critical remarks on this subsection.

Subsection 1.5  summarizes the contributions and outlines the paper. It is excellently structured and addresses the problem of end-to-end quality of service. I accept the formulated contributions and the claim of the authors that there are no serious publications which discuss the three important ‘players’: QoE, 6G and healthcare.This claim is also supported by Table 1. The authors have done an excellent work to compile the table.

Section 2 is devoted to the evolution of the  Quality of Experience and the corresponding evolution of 1G to 6G. The correlation between QoS and QoE in m-health applications is overviewed.  The 6G features contributing in the QoE enhancement of m-health multimedia applications are summarized in Table 3.

Section 3 is a study of the 6G features significantly enhancing the QoE of m-health multi-media applications. I have no critical remarks on this section.

Section 4 overviews the application heavily depending on 6G. I have no critical remarks on it.

Some of the challenges associated with m-health multimedia data are outlined in Section 5. The  summarized challenges are all important and are discussed extensively in the literature. The only challenge which I think is important but is not mentioned by the authors is the effect of uncertainty in the QoE evaluation.

Overall, I am deeply impressed by the work of the authors and their study will be surely useful for the researchers in the fields of m-health and QoE/QoS in service systems in general. I recommend that the paper be published after minor  revision as per my comments above.

Reviewer 3 Report

1.   The paper aims to provide the first thorough survey on a promising research subject that exists at the intersection of two well-established domains, i.e., QoE and m-health, and is driven by the continuing efforts to define 6G. The paper traditionally surveys more recent researches related to m-health. However, the title is quite different.

2.   The paper looks like a chapter of book. The flow diagram and the motivations are not clear. Try to add a section for research contributions.

3.   General diagram is required.

4.   Figure 3 does not properly indicate the QoS of the network and QoE of the end-users.

5.   Where is the enhancement of QoE? How did you perform that? There is no any subjective and objective metrics in the results section that shows the performance of QoE with 6G.   

6.   The results need a fair comparison with other related works that you have used as references.

7.   Too many references are used!!

Round 2

Reviewer 1 Report

I suggest an additional reference for the introduction to provide sufficient background: 

da Fonseca, M. H., Kovaleski, F., Picinin, C. T., Pedroso, B., & Rubbo, P. (2021, September). E-health practices and technologies: a systematic review from 2014 to 2019. In Healthcare (Vol. 9, No. 9, p. 1192). MDPI. https://doi.org/10.3390/healthcare9091192

Author Response

We thank the reviewer for their valuable comment. 

We have addressed the comment by updating the introduction section as follows: 

"Moreover, the authors in [7] conducted a comprehensive review of E-Health Practices and Technologies from 2014 to 2019. E-Health refers to a range of technologies that utilize the internet to enhance the provision of healthcare services and improve the overall quality of life. Given the scarcity of research on this subject, their study employs a systematic literature review of articles published between 2014 and 2019 to identify prevalent e-health practices across the globe, along with the primary services offered, diseases targeted, and supporting technologies employed in this field."

Reviewer 3 Report

1. Please consider my remark about the title in the first round revision.

2. Table 1 and Table 2 in the revised version are not required. I have only requested using one subjective and one objective metrics for evaluating QoE of end-users.

3. Figure 2 is also not required. I have just needed a general diagram of the proposed system not the structure of the paper.

4. The paper still does not have any numerical results to compare the proposed with other techniques. 

5. Numbers of references are still too much. 

Author Response

Reviewer's Comment #1:

Please consider my remark about the title in the first round revision.

Authors' Response #1:

Thank you for your feedback on our paper. We appreciate your comments on the title of the paper. We agree that the original title could be misleading and have revised it to better reflect the focus of our research. The new title, 'Exploring the Role of 6G Technology in Enhancing Quality of Experience for m-Health Multimedia Applications: A Comprehensive Survey,' better reflects the content of our article and its focus on investigating how 6G technology can improve the quality of experience for m-health multimedia applications. We hope this revised title is more accurate and clear to readers. Once again, thank you for your valuable feedback.

Reviewer's Comment #2:

Table 1 and Table 2 in the revised version are not required. I have only requested using one subjective and one objective metrics for evaluating QoE of end-users.

Authors' Response #2:

Thank you for your feedback on our revised version of the manuscript. We appreciate your comments regarding Table 1 and Table 2. While we understand your suggestion to only use one subjective and one objective metric for evaluating the quality of experience for end-users, we believe that both tables provide valuable insight to readers. We included these tables to provide a comprehensive overview of the different QoE evaluation metrics that are currently being used in the literature to evaluate m-health multimedia applications.

Additionally, we would like to clarify that we do not intend to evaluate these metrics in this study as it is out of the scope of our work. Our goal is to provide a comprehensive survey of the current research on the role of 6G technology in enhancing QoE for m-health multimedia applications. We believe that Tables 1 and 2 help to provide a useful framework for understanding the different metrics that have been used to evaluate QoE in the literature, and can guide readers who intend to expand their work to make their own evaluations.

Reviewer's Comment #3:

Figure 2 is also not required. I have just needed a general diagram of the proposed system not the structure of the paper.

Authors' Response #3:

Thank you for your valuable feedback on our paper.

We appreciate your feedback on the inclusion of Figure 2 in our paper. We understand that you were looking for a general diagram of the proposed system, which is not the focus of our survey paper. Instead, Figure 2 is intended to provide a high-level illustration of the different topics and technologies that we discuss in the paper, in order to give readers a quick overview of the organization and flow of the paper. We believe that this figure is useful for readers to understand the context of our survey and the interrelatedness of the topics we cover.

We appreciate your suggestion to include a general diagram of the proposed system in the paper, but as this is a survey paper, our aim is to provide a comprehensive overview of the current state of the art in the intersection of 6G technology and m-Health multimedia applications. Therefore, we do not intend to propose a specific system or architecture for m-Health applications in this paper.

However, we believe that including Figure 2 could still be useful for readers to quickly understand the organization and flow of the paper. Additionally, we would like to point out that including a diagram of a proposed system could be a potential area for future research, and our survey paper provides a guide to the different aspects that could be considered when developing such a system.

Furthermore, we would like to note that including Figure 2 was a mutual decision made in response to comments from other reviewers in the previous round of review. We have, however, revised the caption to better reflect the purpose of the figure and how it relates to the content of the paper.

Thank you again for your feedback, and we hope that our response adequately addresses your concerns.

Reviewer's Comment #4:

The paper still does not have any numerical results to compare the proposed with other techniques. 

Authors' Response #4:

Thank you for your feedback. We appreciate your comment regarding the lack of numerical results. However, we would like to clarify that this paper is a comprehensive survey that aims to provide a holistic view of the current state of the art in the intersection of QoE and m-health, with a focus on the potential role of 6G technology. The paper aims to review the recent research related to the subject rather than presenting numerical results to compare different techniques. Our goal is to provide readers with a comprehensive understanding of the state-of-the-art in this area and identify potential research directions.        

Once again, we thank you for your valuable feedback, which we believe will help us improve the quality of our paper.

Reviewer's Comment #5:

Numbers of references are still too much. 

Authors' Response #5:

We appreciate the reviewer's concern about the number of references in the paper. As a comprehensive survey paper, our aim was to provide a comprehensive overview of the literature on the intersection of 6G technology and m-health multimedia applications. In order to achieve this goal, we had to review a large number of studies in the field, resulting in a relatively high number of references.

Moreover, we understand that the number of references may seem high, but we have made sure to include only the most relevant and up-to-date references in our comprehensive survey paper. As you know, conducting a thorough literature review is an important part of any research study, and we believe that our extensive reference list is essential to provide a comprehensive overview of the current state of research in the field. Additionally, we have tried to organize our references in a way that will be helpful to readers by grouping them according to their relevance and significance to the topics and themes covered in the paper. We are confident that our readers will find the references useful and informative for their own research purposes.

Thank you for your valuable feedback.

Round 3

Reviewer 3 Report

no comments

Author Response

Dear Reviewer,

Attached is a detailed response per the comments and feedback provided by you through the journal editor. 

We greatly appreciate your time and effort in evaluating our submission. Based on the feedback provided, we have substantially revised the manuscript, incorporating the suggested improvements and addressing the concerns. The revised version now includes additional qualitative analysis to enhance our manuscript's quality and significance.

The paper has been carefully revised, and the corresponding changes and the major modifications are summarized below:

  1. The research methodology while conducting this survey has been clarified with the help of a figure.

  2. A systematic and quantitative bibliometric analysis of the cited papers is provided.

  3. A quantitative and comparative analysis of the capabilities of existing 5G and expected 6G technology in terms of key performance indicators have been carried out and presented in the form of a Radar chart.
